# UBP12 and UBP13 negatively regulate the activity of the ubiquitin-dependent peptidases DA1, DAR1 and DAR2

Hannes Vanhaeren[1,2,3,4]*, Ying Chen[1,2], Mattias Vermeersch[1,2], Liesbeth De Milde[1,2], Valerie De Vleeschhauwer[1,2], Annelore Natran[1,2], Geert Persiau[1,2], Dominique Eeckhout[1,2], Geert De Jaeger[1,2], Kris Gevaert[3]*, Dirk Inzé[1,2]*

[1]VIB Center for Plant Systems Biology, Technologiepark, Zwijnaarde, Belgium; [2]Department of Plant Biotechnology and Bioinformatics, Ghent University, Technologiepark, Zwijnaarde, Belgium; [3]VIB Center for Medical Biotechnology, Albert Baertsoenkaai, Ghent, Belgium; [4]Department of Biomolecular Medicine, Ghent University, Albert Baertsoenkaai, Ghent, Belgium

*For correspondence:
hahae@psb.vib-ugent.be (HV);
kris.gevaert@vib-ugent.be (KG);
dirk.inze@psb.vib-ugent.be (DI)

Competing interests: The authors declare that no competing interests exist.

**Abstract** Protein ubiquitination is a very diverse post-translational modification leading to protein degradation or delocalization, or altering protein activity. In *Arabidopsis thaliana*, two E3 ligases, BIG BROTHER (BB) and DA2, activate the latent peptidases DA1, DAR1 and DAR2 by mono-ubiquitination at multiple sites. Subsequently, these activated peptidases destabilize various positive growth regulators. Here, we show that two ubiquitin-specific proteases, UBP12 and UBP13, deubiquitinate DA1, DAR1 and DAR2, hence reducing their peptidase activity. Overexpression of *UBP12* or *UBP13* strongly decreased leaf size and cell area, and resulted in lower ploidy levels. Mutants in which *UBP12* and *UBP13* were downregulated produced smaller leaves that contained fewer and smaller cells. Remarkably, neither UBP12 nor UBP13 were found to be cleavage substrates of the activated DA1. Our results therefore suggest that UBP12 and UBP13 work upstream of DA1, DAR1 and DAR2 to restrict their protease activity and hence fine-tune plant growth and development.

## Introduction

Ubiquitination plays a prominent role in the signaling cascades of many plant hormones (*Santner and Estelle, 2010*), such as auxins (*Salehin et al., 2015*), jasmonates (*Nagels Durand et al., 2016*), gibberellins (*Wang and Deng, 2011*), and strigolactones (*Marzec, 2016*), but also in many plant developmental processes and responses to stress (*Shu and Yang, 2017*). Therefore, a very tight control of this process and a high substrate specificity, which is mainly determined by the E3 ubiquitin ligases (*Shu and Yang, 2017*), are required. The tremendous diversity of the ubiquitination system and its potential in post-translational regulation are illustrated by the presence of more than 1400 genes that encode E3 ligases in Arabidopsis (*Vierstra, 2009*). Furthermore, there is a high diversity of ubiquitination types and combinations with other post-translational modifications (PTMs) (*Callis, 2014*; *Swatek and Komander, 2016*), as well as of the fate of the ubiquitinated protein, such as degradation, delocalization or changes in activity (*Swatek and Komander, 2016*).

In contrast to the more intensively studied action of E3 ligases, insights into the specific roles of deubiquitination enzymes (DUBs) in plant growth and development are only recently emerging. DUBs can generate free ubiquitin from tandem-linear repeats (*Callis et al., 1995*; *Callis et al., 1990*), are able to trim ubiquitin chains by hydrolyzing the isopeptide bond between ubiquitin molecules, and can remove covalently bound ubiquitin from proteins (*Komander et al., 2009*). The

Arabidopsis genome codes for around 50 DUBs. As in yeast and mammals, they can be divided into five classes: the ubiquitin C-terminal hydrolases (UHCs), JAB1/MPN/MOV34 (JAMM) domain DUBs that are zinc metalloproteases, ovarian tumor proteases, the Machado-Josephin domain (MJD) DUBs, and the ubiquitin binding proteins (UBPs), which is the largest group (*Isono and Nagel, 2014*). All UBPs contain specific catalytic Cys- and His-boxes, which are highly conserved in both sequence and length (*Zhou et al., 2017*). Based on their sequence homology and protein domain organization, these 27 members can be further divided into 14 subfamilies (*Yan et al., 2000*). UBP12 and UBP13 are the largest UBPs and contain a unique meprin and TRAF homology (MATH) domain. They were first reported to be functional deubiquitinating enzymes that negatively regulate plant immunity (*Ewan et al., 2011*). Since then, both proteins have been described to be involved in diverse molecular pathways. Mutations in *UBP12* and *UBP13* result in early flowering and a decreased periodicity of circadian rhythm (*Cui et al., 2013*). Molecularly, GIGANTEA (GI) recruits UBP12 and UBP13 to the ZEITLUPE (ZTL) photoreceptor complex, which antagonizes the E3 ligase activity of ZTL and hereby stabilizes GI, ZTL and TOC1 [TIMING OF CAB (CHLOROPHYLL A/B-BIND-ING PROTEIN EXPRESSION) 1] protein levels (*Lee et al., 2019*). In addition, UBP12 and UBP13 can regulate the expression of several genes by deubiquitinating ubiquitinated H2A (H2Aub) upon associating with LIKE HETEROCHROMATIN PROTEIN 1, a plant-specific polycomb group (PcG) protein (*Derkacheva et al., 2016*). Polyubiquitination of MYC2 by the PUB10 E3 ligase can be counteracted by UBP12 and UBP13, preventing degradation of MYC2 by the 26S proteasome which then activates jasmonic acid signaling (*Jeong et al., 2017*). In a similar manner, ROOT GROWTH FACTOR RECEP-TOR 1 (RGFR1) and ORESARA 1 (ORE1) are deubiquitinated and therefore stabilized by UBP12 and UBP13, leading to an increased root sensitivity to the RGF1 peptide hormone (*An et al., 2018*) and an acceleration in leaf senescence (*Park et al., 2019*), respectively.

Mutations in *UBP12* or *UBP13* decrease rosette leaf number and double mutants display severe developmental defects (*Cui et al., 2013*). However, a direct link between these deubiquitinating enzymes and leaf growth and development remains elusive. Here, we found that UBP12 and UBP13 interact with DA1, DAR1 and DAR2 in vivo. DA1, DAR1 and DAR2 have previously been documented to negatively regulate leaf growth. Upon multiple mono-ubiquitination by BIG BROTHER (BB) or DA2, these latent peptidases are activated to cleave growth regulators, such as UBP15, TCP14, TCP15 and TCP22 (*Dong et al., 2017*). In addition, the activating E3 ligases BB and DA2 are cleaved and BB is subsequently degraded by the N-degron pathway, mediated by PROTEOLYSIS 1 (PRT1) (*Dong et al., 2017*). Single knock-outs in *DA1*, *DAR1* and *DAR2* only have very subtle effects on organ size (*Dong et al., 2017*; *Li et al., 2008*). Plant growth is however strongly enhanced in the double mutant *da1ko_dar1-1*, comparable to *da1-1* mutants, which carry a point mutation (DA1$^{R358K}$) (*Li et al., 2008*). The latter mutation has a dominant-negative action towards DA1 and DAR1 (*Li et al., 2008*) and causes a reduction in peptidase activity (*Dong et al., 2017*). Rosette growth is however severely impaired in the triple mutant *da1ko_dar1-1_dar2-1*, and the size of leaf cells and the extent of endoreduplication are reduced (*Peng et al., 2015*). This phenotype can be complemented by ectopic expression of *DA1*, *DAR1* or *DAR2*, suggesting they work redundantly in leaves (*Peng et al., 2015*). On the other hand, overexpression of *GFP-DA1* results in smaller organs with fewer cells (*Vanhaeren et al., 2017*). Mutants of *UBP15* can abolish the *da1-1* phenotype and give rise to smaller organs (*Du et al., 2014*). Inversely, ectopic expression of *UBP15* enhances growth (*Du et al., 2014*; *Liu et al., 2008*).

Here, we demonstrate that UBP12 and UBP13 not only bind DA1, DAR1 and DAR2, but can also remove ubiquitin from these proteins, rendering them in an inactive state. Moreover, UBP12 and UBP13 were not found to be proteolytically cleaved by DA1, DAR1 or DAR2, indicating they work upstream in this pathway. In line with these findings, *UBP12* and *UBP13* mutants and overexpression lines exhibit macroscopic, cellular and molecular phenotypes overlapping with those of *35S::GFP-DA1* overexpression lines and *da1ko_dar1-1_dar2-1* mutants, respectively. Our data provide evidence for a pivotal role of UBP12 and UBP13 in restricting the protease activity of DA1, DAR1 and DAR2 during plant growth and development.

## Results

### DA1, DAR1 and DAR2 interact with UBP12 and UBP13 in vivo

Genetic modifier screens previously identified several interactors of DA1 that either activate its peptidase activity or are subjected to proteolytic cleavage by DA1. To gain further insights into the DA1 growth-regulatory pathway, we generated Arabidopsis lines that overexpressed GFP-tagged fusion proteins of DA1, DAR1 and DAR2. Total protein extracts were isolated from eight-day-old seedlings and incubated with anti-GFP beads to purify the bait proteins and their interactors. Label free quantification identified a significant enrichment (p-value<0.01) of the UBIQUITIN-SPECIFIC PROTEASE 12 (UBP12) and UBP13 in the GFP-DA1 and GFP-DAR1 samples (*Figure 1A–B*) among other interaction candidates (*Figure 1—figure supplement 1*, *Figure 1—source data 1*). Despite the much lower levels of DAR2 MS/MS counts and a lower DAR2 bait protein coverage (*Figure 1—figure supplements 2–3*), we found a significant enrichment of UBP12 at a less stringent threshold (p-value<0.05) and UBP13 at the border of significance (*Figure 1C*, *Figure 1—source data 1*). To confirm this interaction, we performed an in vitro pull-down using UBP12 and UBP13 as bait proteins. DA1, DAR1 and DAR2 were expressed and isolated as HIS-MBP fusion proteins and UBP12 and UBP13 as GST fusion proteins; free GST was used as a negative control. We incubated equal amounts of free GST, full-length GST-UBP12 and GST-UBP13 with HIS-MBP-DA1, HIS-MBP-DAR1 and HIS-MBP DAR2. Western blot analysis after purification with anti-GST beads showed that HIS-MBP-DA1, HIS-MBP-DAR1 and HIS-MBP-DAR2 could be co-purified with either GST-UBP12 or GST-UBP13, but not with free GST (*Figure 1D–F*). Additionally, we co-expressed RFP-DA1, RFP-DAR1 and RFP-DAR2 with either GFP-UBP12 or free GFP from the same vector in Arabidopsis cell suspension cultures. Again, we could confirm the interaction between GFP-UBP12 and RFP-DA1, RFP-DAR1 or RFP-DAR2 (*Figure 1G*).

Next, we measured the expression levels of *DA1*, *DAR1*, *DAR2*, *UBP12* and *UBP13* in isolated wild-type (Col-0) leaves in a detailed time-course from leaf primordium initiation to maturity. Both *UBP12* and *UBP13* are highly and evenly expressed throughout leaf development at comparable levels (*Figure 1—figure supplement 4*). Also *DA1*, *DAR1* and *DAR2* are expressed during leaf development as demonstrated before (*Peng et al., 2015*), albeit *DAR2* at a lower level (*Figure 1—figure supplement 4*). Previous research showed that UBP12 and UBP13 are localized in the nucleus and cytoplasm (*Cui et al., 2013*). Additional transient expression of GFP-UBP12 or GFP-UBP13 in *Nicotiana benthamiana* leaves demonstrated that these proteins co-localize in the cytoplasm and the nucleus with RFP-DA1, RFP-DAR1 and RFP-DAR2 (*Figure 1—figure supplements 5–10*). Taken together, UBP12 and UBP13 co-localize with DA1, DAR1 and DAR2, are co-expressed during leaf development and interact with DA1, DAR1 and DAR2 in vivo and in vitro.

### Miss-expression of *UBP12* and *UBP13* alters leaf size

Because DA1, DAR1 and DAR2 are known to restrict plant organ growth, we examined the role of UBP12 and UBP13 in regulating leaf size by generating several independent transgenic *35S::UBP12* and *35S::UBP13* lines. All overexpression lines showed a reduction in rosette area (*Figure 2A*). In addition, leaves appeared to be rounder than those of Col-0, a phenotype that can also be observed in *da1-1* plants, and the petioles were found to be shorter. To quantify this, we measured the leaf blade height and width of Col-0, *da1-1*, *35S::UBP12* and *35S::UBP13* plants and calculated the leaf blade index, which is the ratio of leaf height to width and hence a measure of leaf shape (*Tsukaya, 2002*). We could observe a significantly lower leaf blade index in *da1-1* mutants (*Figure 2—figure supplement 1*) and in two independent *35S::UBP12* and *35S::UBP13* lines compared to Col-0 (*Figure 2—figure supplement 2*), meaning the leaves had a rounder shape. For several *35S::UBP13* lines, we were unable to produce stable seed stocks because the homozygous plants were stunted in growth (*Figure 2A*) and failed to produce a flower stalk and seeds. We continued with two independent lines for further phenotypic analysis: two homozygous *UBP12* lines (*35S::UBP12_3.1* and *35S::UBP12_3.2*) and two *UBP13* lines from which heterozygous plants were selected (*35S::UBP13_1.1* and *35S::UBP13_2.3*) (*Figure 2A*). These lines all had a significant increase in their respective transgene expression compared to Col-0 (*Figure 2—figure supplement 3*).

In parallel, we screened two independent *UBP12* T-DNA insertion lines [*ubp12-1* (GABI_244E11) and *ubp12-2* (GABI_742C10)] and three independent *UBP13* T-DNA lines [*ubp13-1* (SALK_128312),

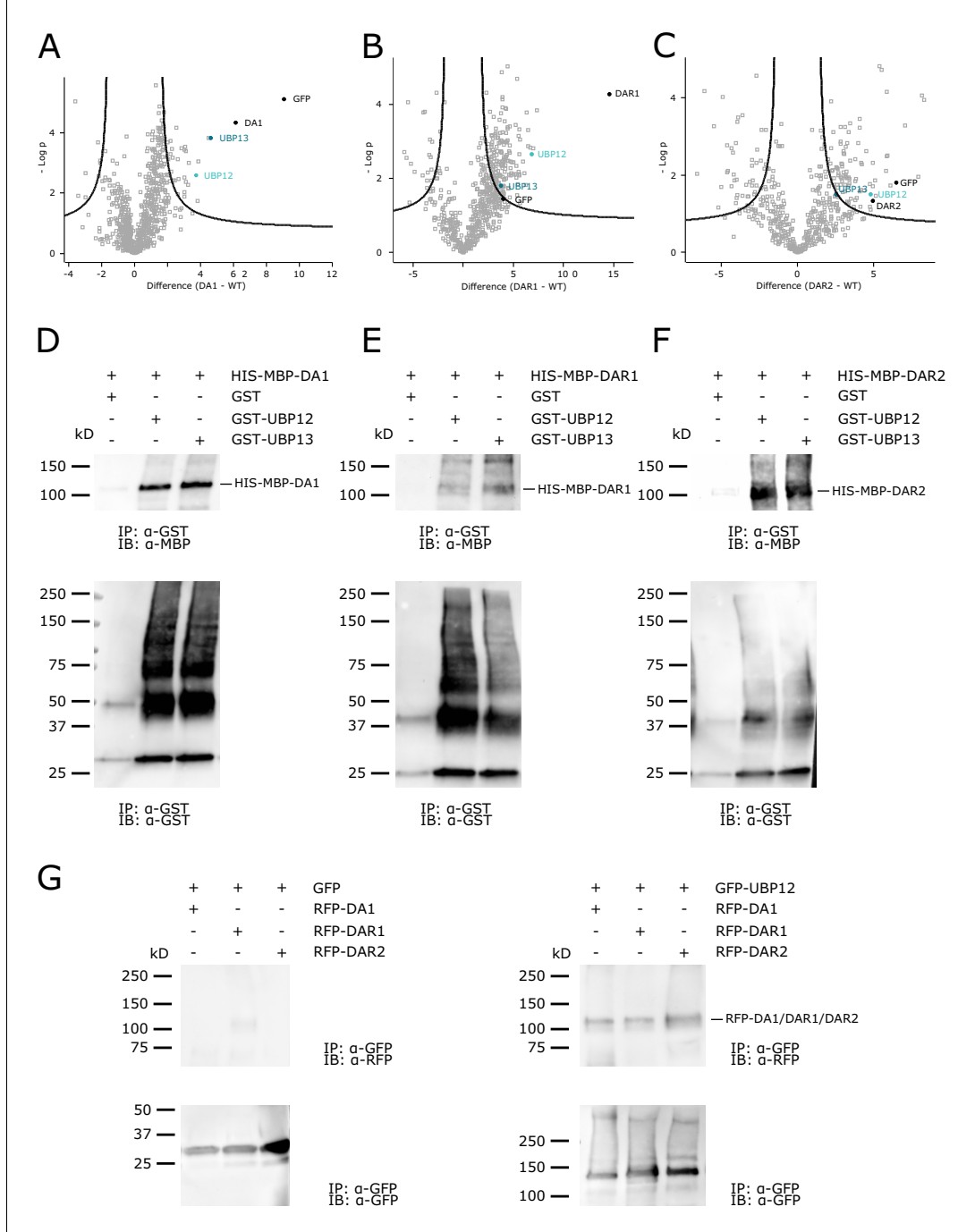

**Figure 1.** UBP12 and UBP13 interact with DA1, DAR1 and DAR2. (**A–C**) Enrichment of the bait, UBP12 and UBP13 compared to the control after immunoprecipitation in (**A**) *35S::GFP-DA1* (FDR = 0.01, S0 = 1, permutation-based FDR-corrected t-test, *Figure 1—source data 1*), (**B**) *35S::GFP-DAR1* (FDR = 0.01, S0 = 1, permutation-based FDR-corrected t-test, *Figure 1—source data 1*) or (**C**) *35S::GFP-DAR2* (FDR = 0.05, S0 = 1, permutation-based FDR-corrected t-test, *Figure 1—source data 1*) seedlings. (**D–F**) In vitro pull down of (**D**) HIS-MBP-DA1, (**E**) HIS-MBP-DAR1 and (**F**) HIS-MBP-DAR2 with free GST, GST-UBP12 and GST-UBP13. (**G**) In vivo pull-down of RFP-DA1, RFP-DAR1 and RFP-DAR2 with free GFP and GFP-UBP12.

The online version of this article includes the following source data and figure supplement(s) for figure 1:

**Source data 1.** List of DA1, DAR1, DAR2 interactors and LFQ intensities by MS/MS.

**Source data 2.** MS/MS counts of DA1, DAR1 and DAR2; protein coverage of DA1, DAR1 and DAR2; relative expression levels of *DA1*, *DAR1*, *DAR2*, *UBP12* and *UBP13* during leaf development.

**Figure supplement 1.** Overlap between potential interactors of DA1, DAR1 and DAR2.

*Figure 1 continued on next page*

*Figure 1 continued*

**Figure supplement 2.** MS/MS counts of DA1, DAR1 and DAR2 baits from GFP-DA1, GFP-DAR1 and GFP-DAR2 pull-downs, respectively (Bars represent the SEM; n = 3 biological repeats; *Figure 1—source data 2*).

**Figure supplement 3.** Percentage coverage of the DA1, DAR1 and DAR2 baits from GFP-DA1, GFP-DAR1 and GFP-DAR2 pull-downs, respectively (Bars represent the SEM; n = 3 biological repeats; *Figure 1—source data 2*).

**Figure supplement 4.** Relative expression levels of *UBP12*, *UBP13*, *DA1*, *DAR1* and *DAR2* throughout leaf development, scaled to the lowest value (DAR2, 5 DAS; *Figure 1—source data 2*).

**Figure supplement 5.** Co-localization of GFP-UBP12 and RFP-DA1 in *Nicotiana benthamiana* leaves.

**Figure supplement 6.** Co-localization of GFP-UBP13 and RFP-DA1 in *Nicotiana benthamiana* leaves.

**Figure supplement 7.** Co-localization of GFP-UBP12 and RFP-DAR1 in *Nicotiana benthamiana* leaves.

**Figure supplement 8.** Co-localization of GFP-UBP13 and RFP-DAR1 in *Nicotiana benthamiana* leaves.

**Figure supplement 9.** Co-localization of GFP-UBP12 and RFP-DAR2 in *Nicotiana benthamiana* leaves.

**Figure supplement 10.** Co-localization of GFP-UBP13 and RFP-DAR2 in *Nicotiana benthamiana* leaves.

*ubp13-2* (SALK_024054) and *ubp13-3* (SALK_132368)]. After leaf area measurements, we could observe a decrease in leaf area in the *ubp12-2* mutants (*Figure 2B*), in which the levels of both *UBP12* and *UBP13* transcripts were previously shown to be downregulated (*Cui et al., 2013*). Mutant lines in which only the expression of either *UBP12* or *UBP13* was downregulated, displayed no differences in leaf size compared to the control (*Figure 2—figure supplements 4–7*).

The final leaf size is determined by cell proliferation and cell expansion. To identify which of these processes were affected in the *ubp12-2* mutant and in the *UBP12* and *UBP13* overexpression lines, we performed a cellular analysis on the abaxial pavement cells of mature leaves. In all overexpression lines, the significant decrease in leaf area (29%, 26%, 33% and 36% for *35S::UBP12_3.1*, *35S::UBP12_3.2*, *35S::UBP13_1.1* and *35S::UBP13_2.3*, respectively) was caused by a strong reduction in cell area (26%, 40%, 24% and 46% for *35S::UBP12_3.1*, *35S::UBP12_3.2*, *35S::UBP13_1.1* and *35S::UBP13_2.3*, respectively), whereas the decrease in *ubp12-2* leaf size (32%) resulted from a reduction in cell area (21%) and cell number (11%) (*Figure 2C*). Remarkably, besides a general decrease in pavement cell area, we could observe a larger proportion of very small cells in the *UBP12* and *UBP13* overexpression lines (*Figure 2—figure supplement 8*), which was even more pronounced in homozygous *35S::UBP13_1.1* plants (*Figure 2—figure supplement 9*). A cell area distribution plot confirmed that indeed all independent *UBP12* and *UBP13* overexpression lines harbored a larger proportion of these small cells in addition to a general decrease in mature pavement cell size (*Figure 2D*). For the *ubp12-2* mutant, no differences in cell area distribution compared to Col-0 could be observed (*Figure 2—figure supplement 10*).

## Overexpression of *UBP12* or *UBP13* delays the onset of endoreduplication and cell differentiation in leaves

A strong reduction in cell size is often correlated with decreased levels of endoreduplication. To explore this into more detail, we harvested a time-course of the first leaf pair of Col-0, *35S::UBP12_3.1* and *35S::UBP13_1.1* (hereafter referred to as *35S::UBP12* and *35S::UBP13*) plants, spanning all major developmental time points. At nine days after stratification (DAS), all leaf cells of the three lines exhibited a 2C or 4C content, demonstrating the majority of cells are still in the mitotic cell cycle (*Figure 3A*). From 12 DAS onwards, cells with 8C started to appear, indicating the onset of endoreduplication. At 12 DAS, leaves of *35S::UBP12* plants contained a significantly larger proportion of 2 C cells (53%) than those of the Col-0 (43%) and a significantly lower amount of 4 C cells (44% and 52%, respectively). At 15 DAS, the amount of 8C nuclei was significantly lower in *35S::UBP12* (28%) than in the control (35%) (*Figure 3A*). Similar, but more pronounced observations were found in *35S::UBP13* leaves, in which a significantly higher amount of 2 C cells was detected at 12 DAS (61%) and 15 DAS (42%) compared to Col-0 (43% and 25%, respectively). In addition, a lower level of 4 C cells was found at 12 DAS (35% in *35S::UBP13*, 52% in Col-0) and fewer cells with 8C were present at 15, 18, 21 and 27 DAS in *35S::UBP13* (22%, 33%, 35% and 35%, respectively) compared to Col-0 (35%, 42%, 42% and 46%, respectively) (*Figure 3A*). An alternative way to illustrate endoreduplication levels is the endoreduplication index, representing the average amount of endocycles a nucleus underwent. Generally, slightly lower endoreduplication levels could be observed in

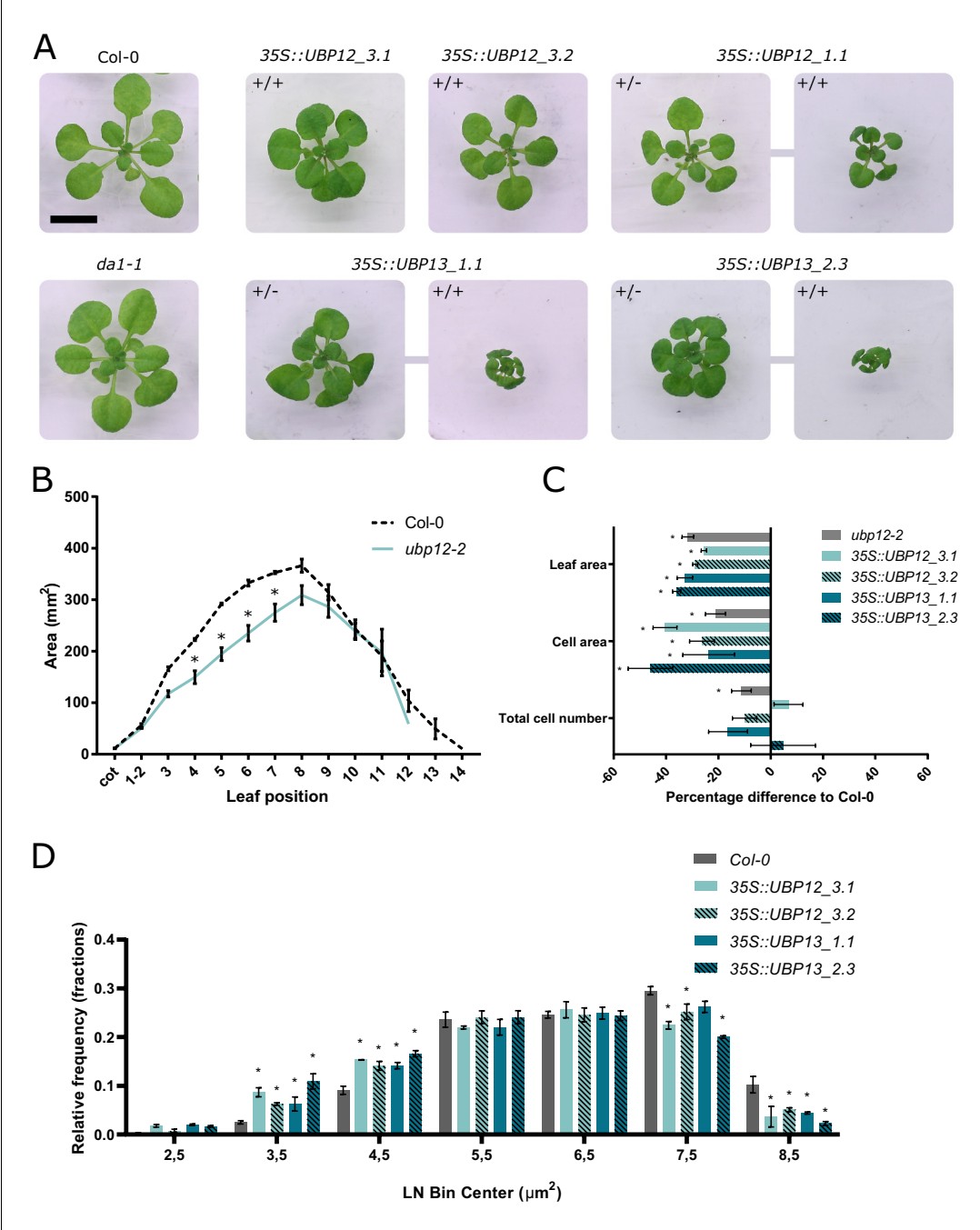

**Figure 2.** Regulation of leaf size by UBP12 and UBP13. (**A**) Twenty-one-day-old plants of Col-0, *da1-1* and homozygous (+/+) or heterozygous (+/-) *UBP12* and *UBP13* overexpression lines. Scale bar represents 1 cm, homozygous and heterozygous plants of the same single locus line are linked in the figure. (**B**) Leaf area measurements of Col-0 and *ubp12-2*, n = 3 biological repeats with >10 plants per repeat. (**C**) Percentage differences of leaf area, cell area and cell number of *ubp12-2*, *35S::UBP12_3.1*, *35S::UBP12_3.2*, *35S::UBP13_1.1* and *35S::UBP13_3.2* compared to Col-0, n = 3 biological repeats with three representative leaves per repeat. (**D**) Relative frequencies of LN transformed cell area distribution of Col-0, *35S::UBP12_3.1*, *35S::UBP12_3.2*, *35S::UBP13_1.1* and *35S::UBP13_3.2* in 1-$\mu$m$^2$ bin sizes, n = 3 biological repeats with three representative leaves per repeat. Bars represent the SEM; * indicates p-value<0.05, ANOVA (*Figure 2—source data 1*).

The online version of this article includes the following source data and figure supplement(s) for figure 2:

**Source data 1.** Leaf area analysis and statistics of Col-0 and *ubp12-2*; leaf area analysis and statistics of Col-0, *35S::UBP12_3.1*, *35S::UBP12_3.2*, *35S::UBP13_1.1* and *35S::UBP13_3.2*; cellular analysis and statistics of Col-0, *35S::UBP12_3.1*, *35S::UBP12_3.2*, *35S::UBP13_1.1* and *35S::UBP13_3.2*; cellular analysis of *ubp12-2*, relative frequency of the pavement cell area and statistics of Col-0, *35S::UBP12_3.1*, *35S::UBP12_3.2*, *35S::UBP13_1.1* and *35S:: UBP13_3.2*.

*Figure 2 continued on next page*

*Figure 2 continued*

**Source data 2.** Q-RT-PCR data and statistics of *UBP12* and *UBP13* expression in Col-0, *da1-1*, *35S::UBP12_3.1*, *35S::UBP12_3.2*, *35S::UBP13_1.1* and *35S::UBP13_3.2* overexpression lines; leaf area data and statistics of *ubp12-1*; leaf area data and statistics of *ubp13-1*; leaf area data and statistics of *ubp13-2*; leaf area data and statistics of *ubp13-3*; relative frequency of *ubp12-2* pavement cell area data and statistics.
**Figure supplement 1.** Leaf index measurements of Col-0 and *da1-1* mutants, n = 3 biological repeats with >6 leaves per repeat.
**Figure supplement 2.** Leaf index measurements of Col-0, *35S::UBP12_3.1*, *35S::UBP12_3.2*, *35S::UBP13_1.1* and *35S::UBP13_3.2* overexpression lines.
**Figure supplement 3.** *UBP12* and *UBP13* transcript levels in Col-0, *da1-1*, *35S::UBP12_3.1*, *35S::UBP12_3.2*, *35S::UBP13_1.1* and *35S::UBP13_3.2* overexpression lines, n = 3 biological repeats with >8 leaves per repeat.
**Figure supplement 4.** Leaf area measurements of *ubp12-1*, compared to Col-0, n = 3 biological repeats with >10 plants per repeat.
**Figure supplement 5.** Leaf area measurements of *ubp13-1*, compared to Col-0, n = 3 biological repeats with >10 plants per repeat.
**Figure supplement 6.** Leaf area measurements of *ubp13-2* and compared to Col-0, n = 3 biological repeats with >10 plants per repeat.
**Figure supplement 7.** Leaf area measurements of *ubp13-3* compared to Col-0, n = 3 biological repeats with >10 plants per repeat.
**Figure supplement 8.** Representations of abaxial leaf epidermal cells of Col-0, homozygous *35S::UBP12_3.1*, homozygous *35S::UBP12_3.2*, heterozygous *35S::UBP13_1.1* and heterozygous *35S::UBP13_3.2*.
**Figure supplement 9.** Representations of abaxial leaf epidermal cells of Col-0 and the homozygous *35S::UBP13_1.1*, (+/+) indicates homozygous plants, scale-bar represents 0.1 mm.
**Figure supplement 10.** Cell size distribution plot of Col-0 and *ubp12-2*, n = 3 biological repeats with three representative leaves per repeat.

*35S::UBP12* leaves (*Figure 3—figure supplement 1*), whereas a stronger effect was clear in *35S::UBP13* leaves, with significant differences at 12, 15 and 21 DAS (*Figure 3—figure supplement 2*).

Because the strongest effects were visible in the younger time points, we subsequently prepared RNA from the younger third leaf until 18 DAS. Considering the higher 2C levels in *35S::UBP12* and *35S::UBP13* (*Figure 3A*) and the presence of larger proportions of small cells in the leaf epidermis at 21 DAS (*Figure 2D*, *Figure 2—figure supplements 8–9*), we measured the expression of several genes that are highly expressed during cell proliferation and whose expression goes down during the transition to differentiation and endoreduplication in leaves: *ANGUSTIFOLIA3* (*AN3*) (*Horiguchi et al., 2005*), *AINTEGUMENTA* (*ANT*), (*Mizukami and Fischer, 2000*), *GROWTH-REGU-LATING FACTOR 5* (*GRF5*) (*Horiguchi et al., 2005*; *Vercruyssen et al., 2015*), *KLUH* (*Anastasiou et al., 2007*) and the cell division marker *CYCLINB1;1* (*CYCB1;1*) (*Doerner et al., 1996*; *Shaul et al., 1996*). At 9 DAS, when all epidermal cells are actively dividing (*Andriankaja et al., 2012*), we could observe a significantly higher expression of *ANT*, *GRF5* and *KLUH* in *35S::UBP13* lines compared to the Col-0 (*Figure 3B*). Later in development at 12 DAS, a large portion of the Col-0 cells should have exited the mitotic division cycle and start to differentiate (*Andriankaja et al., 2012*; *Donnelly et al., 1999*; *Kazama et al., 2010*). Still, there was a significant higher expression of *AN3* and *GRF5* and the cell division marker *CYCB1;1* in *35S::UBP13* leaves (*Figure 3C*). On 15 DAS, the expression levels of *ANT* and *CYCB1;one* were elevated in *35S::UBP13* leaves (*Figure 3—figure supplement 3*) and finally, at 18 DAS, *ANT* was still significantly higher expressed in *35S::UBP13* compared to the Col-0 (*Figure 3—figure supplement 4*). To further explore these observations, we characterized the epidermal cells of tips of the third leaf of 12 day old seedlings, which should be undergoing cell differentiation at this stage in Col-0 (*Andriankaja et al., 2012*). In Col-0 leaf tips, we could indeed observe that the epidermal cells had started to expand and differentiate (*Figure 3D*). On the contrary, similar leaf tips of *35S::UBP12* or *35S::UBP13* plants contained almost exclusively undifferentiated cells (*Figure 3D*). These results, together with our observation that cell number was not significantly altered in *UBP12* an *UBP13* overexpression lines, indicate that especially during the early developmental stages of leaf growth, endoreduplication and cell differentiation are delayed in *35S::UBP12* and *35S::UBP13* leaves.

## UBP12 and UBP13 can deubiquitinate activated DA1, DAR1 and DAR2

The latent peptidases DA1, DAR1 and DAR2 can be activated upon ubiquitination by the E3 ligases BB or DA2. Considering the enzymatic function of UBP12 and UBP13, they might counteract this by deubiquitinating these activated peptidases. To test this, we performed an in vitro ubiquitination assay to generate ubiquitinated DA1, DAR1 an DAR2, followed by a deubiquitination step with UBP12 or UBP13, or their respective catalytic mutants, UBP12$^{C208S}$ and UBP13$^{C207S}$ (*Cui et al., 2013*; *Ewan et al., 2011*). DA1, DAR1 and DAR2 were expressed and isolated as HIS-MBP fusion proteins and UBP12, UBP13, UBP12$^{C208S}$ and UBP13$^{C207S}$ as GST fusion proteins. To mediate the

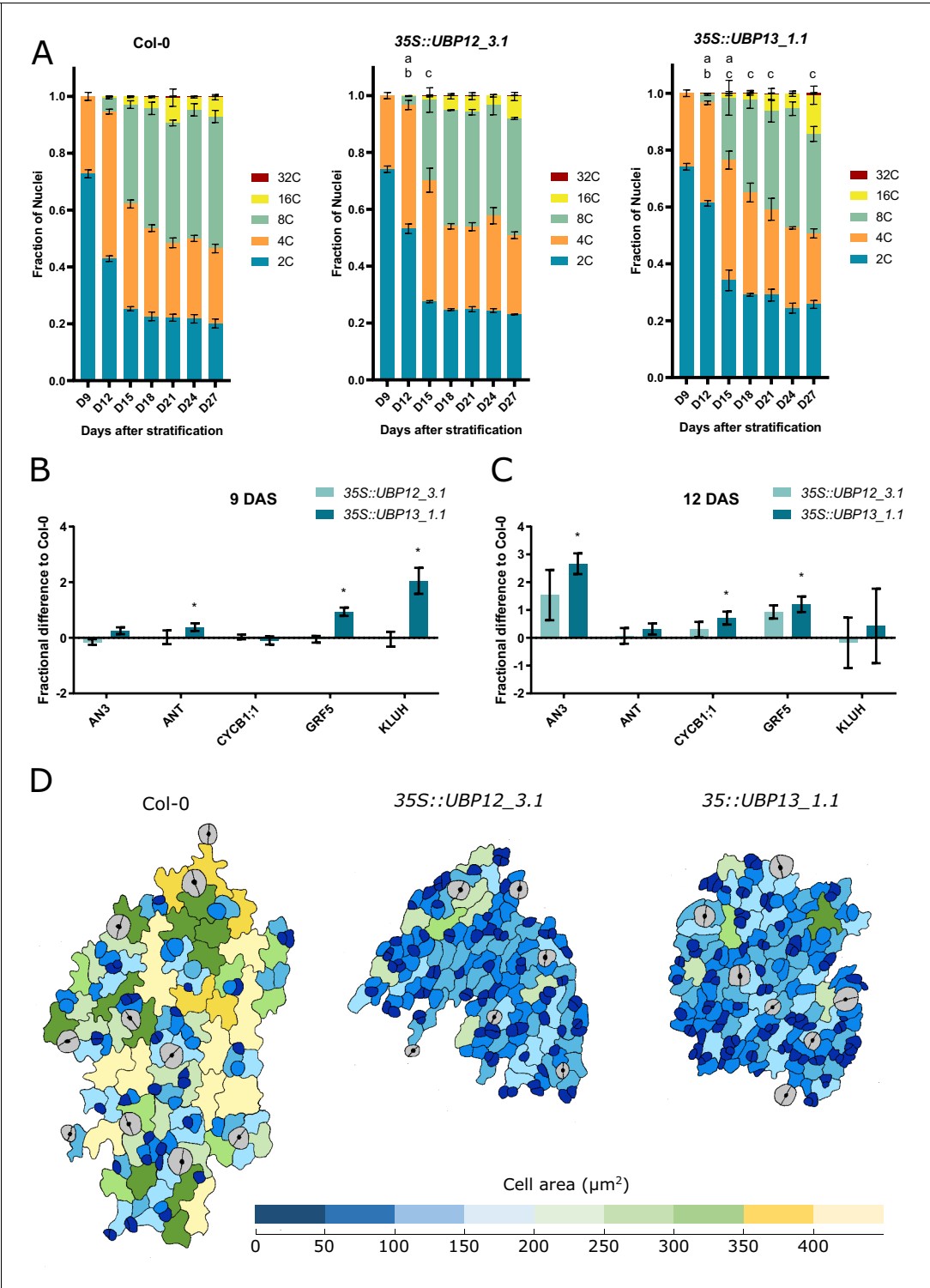

**Figure 3.** UBP12 and UBP13 regulate the onset of endoreduplication. (A) Ploidy distribution of nuclear DNA in Col-0, *35S::UBP12_3.1* and *35S::UBP13_1.1*. (B–C) Fractional difference in expression of cell proliferation markers in *35S::UBP12_3.1* and *35S::UBP13_1.1* compared to Col-0 at (B) nine and (C) 12 DAS. (D) Representations of abaxial leaf epidermal cells at the tip of the third leaf of Col-0, *35S::UBP12_3.1* and *35S::UBP13_1.1* at 12 DAS. Bars represent the SEM, n = 3 biological repeats with >3 leaves per repeat; a, b and c indicate a significant difference in 2C, 4C and 8C, respectively; * indicates p-value<0.05, ANOVA, (*Figure 3—source data 1*).

The online version of this article includes the following source data and figure supplement(s) for figure 3:

**Source data 1.** Flow cytometry counts and statistics; Q-RT-PCR data and statistics of proliferation markers in developing leaves (9 DAS and 12 DAS).

*Figure 3 continued on next page*

**Source data 2.** Endoreduplication index calculations and statistics; Q-RT-PCR data and statistics of proliferation markers in developing leaves (15 DAS and 18 DAS).
**Figure supplement 1.** Endoreduplication index of *35S::UBP12_3.1* leaf nuclei compared to Col-0.
**Figure supplement 2.** Endoreduplication index of *35S::UBP13_1.1* leaf nuclei compared to Col-0.
**Figure supplement 3.** Relative expression levels of cell proliferation markers in Col-0, *35S::UBP12_3.1* and *35S::UBP13_1.1* at 15 DAS.
**Figure supplement 4.** Relative expression levels of cell proliferation markers in Col-0, *35S::UBP12_3.1* and *35S::UBP13_1.1* at 18 DAS.

ubiquitination, recombinant HIS-DA2 was purified as the E3 ligase. The ubiquitinated peptidases were incubated with equal amounts of either GST-UBP12, GST-UBP12$^{C208S}$, GST-UBP13 or GST-UBP13$^{C207S}$. *Figure 4A* shows that ubiquitinated HIS-MBP-DA1 can be deubiquitinated by GST-UBP12 or GST-UBP13, but their respective catalytic mutants fail to do so. Similar deubiquitination activities could be observed for the substrates DAR1 and DAR2 (*Figure 4B–C*). Because the peptidase activity of DA1, DAR1 and DAR2 is similarly activated upon ubiquitination and they function redundantly to regulate leaf size and ploidy, we chose to focus on DA1 in our next experiments.

To demonstrate the specificity of UBP12 and UBP13, we performed additional deubiquitination experiments with several UBPs from different subfamilies (*Yan et al., 2000*): GST-UBP3, GST-UBP24 and GST-UBP15, of which the latter had already been demonstrated to function in the same pathway of DA1 (*Du et al., 2014*) as its cleavage substrate (*Dong et al., 2017*). Ubiquitinated HIS-MBP-DA1

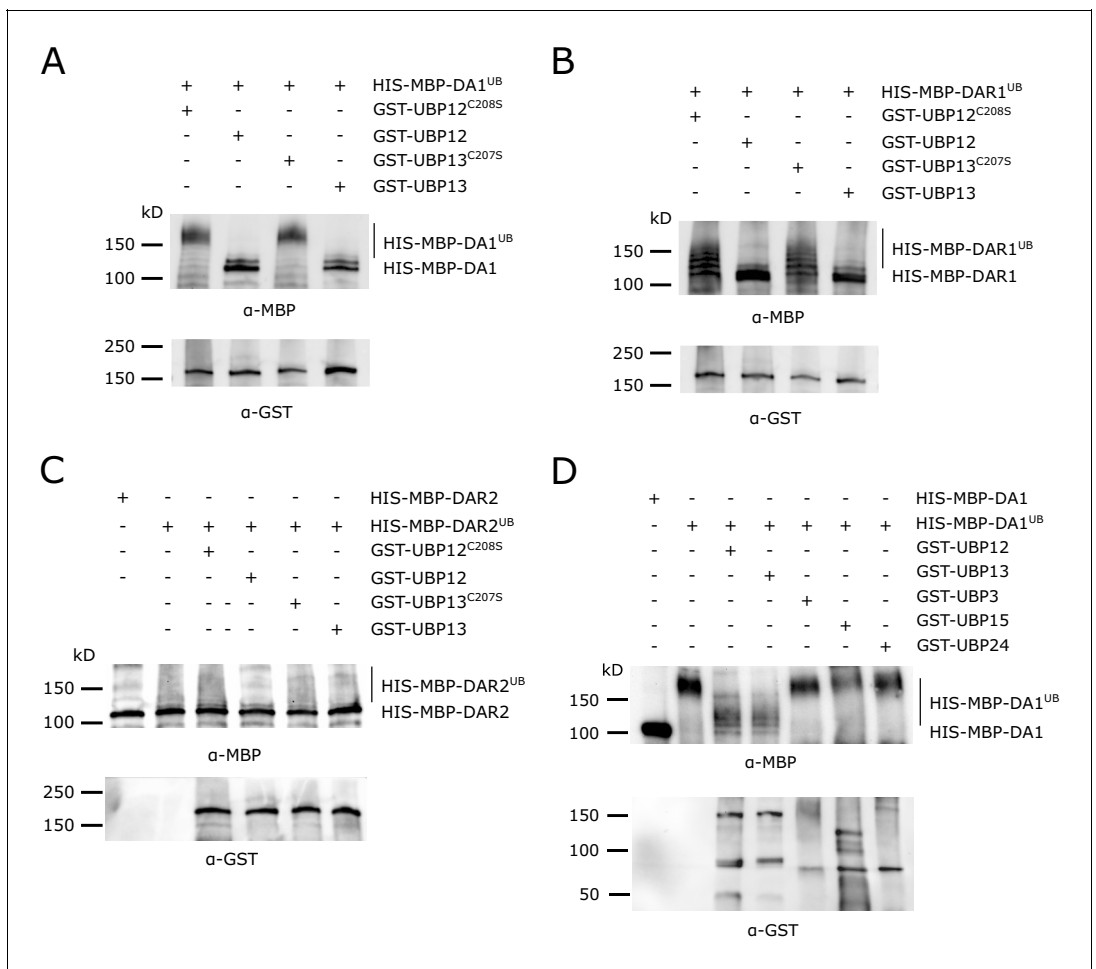

**Figure 4.** In vitro deubiquitination specificity of DA1, DAR1 and DAR2 by UBP12 and UBP13. (A–C) In vitro deubiquitination assays with GST-UBP12, GST-UBP12$^{C208S}$, GST-UBP13 or GST-UBP13$^{C207S}$ of (A) HIS-MBP-DA1, (B) HIS-MBP-DAR1 and (C) HIS-MBP-DAR2. (D) Deubiquitination assay with GST-UBP12, GST-UBP13, GST-UBP3, GST-UBP15 and GST-UBP24 of HIS-MBP-DA1.

was incubated with equal amounts of GST-UBP12, GST-UBP13, GST-UBP3, GST-UBP15 and GST-UBP24. We found again that both GST-UBP12 and GST-UBP13 could strongly deubiquitinate HIS-MBP-DA1, but no such effect could be observed upon addition of GST-UBP3, GST-UBP15 or GST-UBP24 (*Figure 4D*).

To further consolidate these observations, we evaluated the effect of different in vivo levels of UBP12 and UBP13 on the ubiquitination status of DA1. To realize this, we produced ubiquitinated DA1 in vitro and incubated it with equal amounts of cell-free extracts of Col-0 and *ubp12-2* seedlings, the latter containing lower protein levels of UBP12 and UBP13. Already after 1 hr, we could observe a shift in molecular weight in the ubiquitinated HIS-MBP-DA1 samples that were incubated with Col-0 extract (*Figure 5A*), and this shift did not occur in the sample that was incubated with the *ubp12-2* extract (*Figure 5B*). In addition, the intensity of ubiquitinated HIS-MBP-DA1 bands faded faster in the presence of Col-0 extract than in the presence of *ubp12-2* extract (*Figure 5A–C*). We performed a similar experiment with extracts of *35S::UBP12* and *35S::UBP13* plants, which contain higher levels of UBP12 or UBP13 proteins respectively. First, we could confirm the faster loss in molecular weight and intensity of ubiquitinated HIS-MBP-DA1 in Col-0 (*Figure 5D*) than in *ubp12-2* after 2 hr and 4 hr of incubation (*Figure 5E*). In addition, when ubiquitinated DA1 proteins were incubated with equal extracts of *35S::UBP12* (*Figure 5F*) or *35S::UBP13* (*Figure 5G*) plants, the intensity of ubiquitinated HIS-MBP-DA1 disappeared at a faster rate (*Figure 5D–H*). At 6 hr or 8 hr of incubation, no HIS-MBP-DA1 bands could be distinguished anymore from the background, these results could therefore not be interpreted (*Figure 5D–H*). In addition, we extracted proteins of *35S::GFP-DA1*, *35S::GFP-DA1_35S::UBP12* and *35S::GFP-DA1_35S::UBP13* plants and performed a purification with anti-GFP beads from equal protein inputs. After purification, we submitted the samples to Western Blot and detected the abundance of GFP-DA1 proteins. We could not identify increases in protein abundance in the *35S::GFP-DA1_35S::UBP12* and *35S::GFP-DA1_35S::UBP13* samples

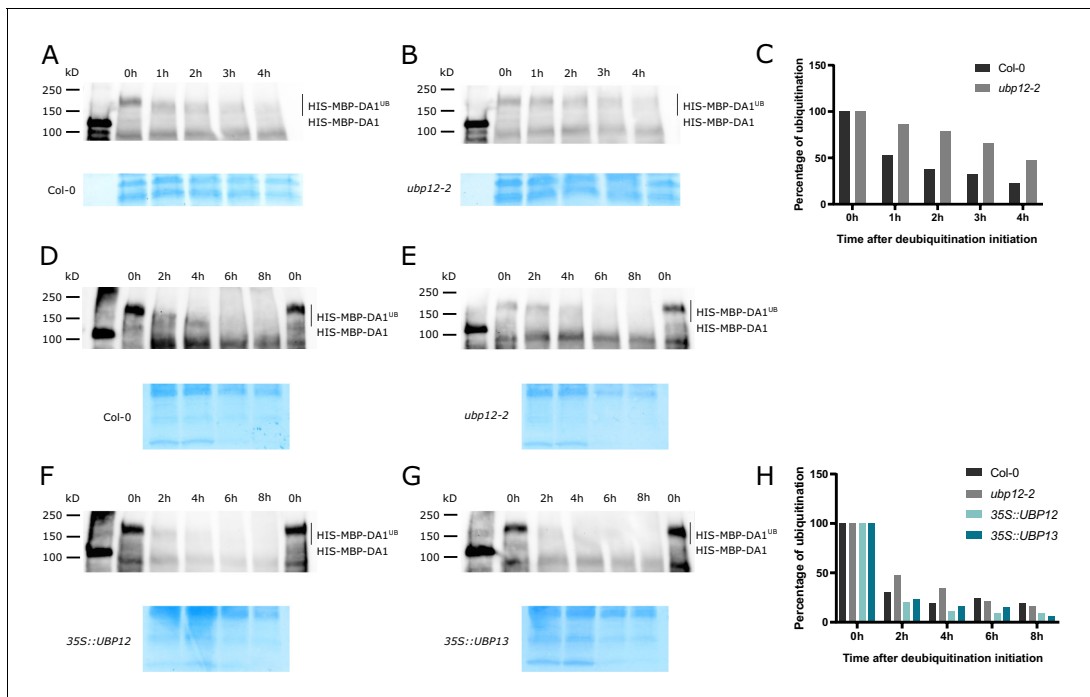

**Figure 5.** Deubiquitination of DA1, DAR1 and DAR2 by UBP12 and UBP13 in vivo. (**A–B**) Cell-free deubiquitination assay of ubiquitinated HIS-MBP-DA1 proteins incubated with (**A**) Col-0 or (**B**) *ubp12-2* extract. (**C**) Quantification of deubiquitination (*Figure 5—source data 1*). (**D–G**) Cell-free deubiquitination assay of ubiquitinated HIS-MBP-DA1 with (**D**) Col-0, (**E**) *ubp12-2*, (**F**) *35S::UBP12* or (**G**) *35S::UBP13* extracts. (**H**) Quantification of deubiquitination (*Figure 5—source data 1*).

The online version of this article includes the following source data and figure supplement(s) for figure 5:

**Source data 1.** Calculation of in vivo deubiquitination.

**Figure supplement 1.** Stability of DA1 in 35::GFP-DA1, 35::GFP-DA1_35::UBP12 and 35S::GFP-DA1_35S::UBP13 seedlings.

compared to those of *35S::GFP-DA1* (*Figure 5—figure supplement 1*), suggesting UBP12 and UBP13 do not prevent potential removal of polyubiquitination of DA1 and hereby prevent its degradation by the proteasome.

These experiments demonstrate the deubiquitination specificity by UBP12 and UBP13 in vitro and the positive effects of high in vivo UBP12 or UBP13 protein levels on the deubiquitination of DA1. In addition, high levels of UBP12 or UBP13 do not seem to affect the stability of DA1 in vivo.

## UBP12 and UBP13 are not substrates of the activated DA1

Our observation that UBP12 and UBP13 can deubiquitinate DA1, DAR1 and DAR2 suggests they work upstream in this growth-regulatory pathway. Considering the peptidase activity of DA1, DAR1 and DAR2, we tested whether these deubiquitinating enzymes could in their turn be substrates. For this purpose, we incubated GST-UBP12, GST-UBP12$^{C208S}$, GST-UBP13 and GST-UBP13$^{C207S}$ with ubiquitinated HIS-MBP-DA1 or the peptidase-deficient HIS-MBP-DA1$^{H418A,H422A}$ (*Dong et al., 2017*). The catalytic UBP mutants were added to the assay because they are unable to deubiquitinate DA1 (*Figure 4A*) and, hence, are exposed for a longer time to the activated peptidase. However, after 4 hr of incubation, the intensities of all GST-tagged UBP proteins were equal and no additional cleaving fragments could be observed in the HIS-MBP-DA1 samples compared to those with HIS-MBP-DA1$^{H418A,H422A}$ (*Figure 6A*). Similar results were observed in the reactions in which the catalytic mutants of UBP12 or UBP13 were incubated with ubiquitinated HIS-MBP-DA1 or HIS-MBP-DA1$^{H418A,H422A}$ (*Figure 6A*).

Previously, it has been demonstrated that the dwarfed phenotype of strong *BB* overexpression lines could largely be rescued by ectopic co-expression of *DA1*, as a result of the cleavage and destabilization of BB proteins (*Dong et al., 2017*). Similarly, we generated double overexpression lines of *35S::GFP-DA1* and *35S::UBP12* or *35S::UBP13* (*Figure 6—figure supplement 1*). Compared to Col-0, leaf areas were reduced in *35S::GFP-DA1* (*Figure 6B*), as described before (*Vanhaeren et al., 2017*). In all double overexpression lines, we could observe similar phenotypes as in the *35S::UBP12* and *35S::UBP13* lines in the Col-0 background (*Figure 6B*, *Figure 2A*), suggesting that the *UBP12* and *UBP13* overexpression phenotype is dominant and UBP12 and UBP13 are not destabilized by DA1.

Thus, our biochemical, genetic and phenotypic analyses show that UBP12 and UBP13 are not cleavage substrates, but act upstream of DA1, DAR1 and DAR2 to counteract their ubiquitination.

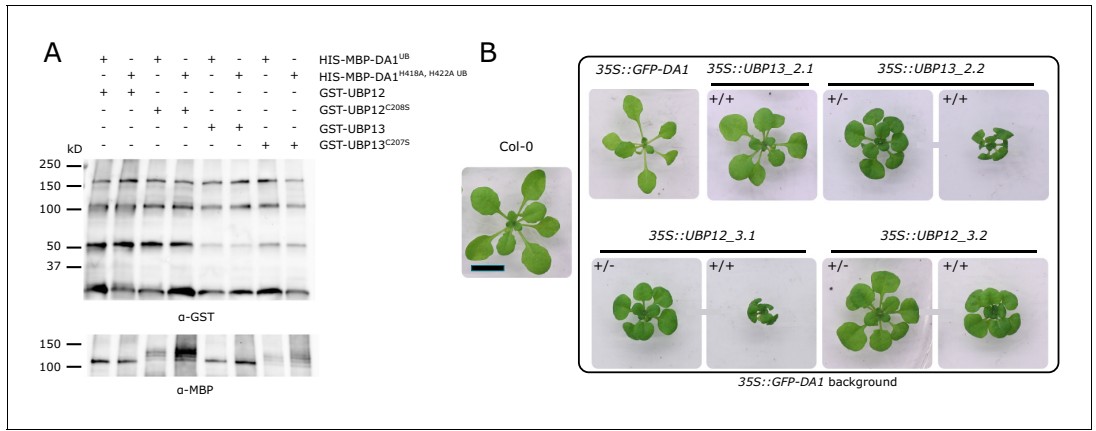

**Figure 6.** Cleaving assay on UBP12 and UBP13. (**A**) In vitro cleaving assay on GST-UBP12, GST-UBP12$^{C208S}$, GST-UBP13 and GST-UBP13$^{C207S}$ by HIS-MBP-DA1 and the peptidase-deficient HIS-MBP-DA1$^{H418A,H422A}$. (**B**) Twenty-one-day-old plants of Col-0, *35S::GFP-DA1*, *35S::UBP12_3.1/35S::GFP-DA1*, *35S::UBP12_3.2/35S::GFP-DA1*, *35S::UBP13_2.1/35S::GFP-DA1* and *35S::UBP13_2.2/35S::GFP-DA1* homozygous (+/+) and heterozygous (+/-) double overexpression lines. Scale bar represents 1 cm.

The online version of this article includes the following source data and figure supplement(s) for figure 6:

**Source data 1.** Q-RT-PCR data and statistics of *DA1*, *UBP12* and *UBP13* expression in Col-0, *35S::GFP-DA1*, *35S::UBP12_3.1/35S::GFP-DA1*, *35S::UBP12_3.2/35S::GFP-DA1*, *35S::UBP13_2.1/35S::GFP-DA1* and *35S::UBP13_2.2/35S::GFP-DA1*.

**Figure supplement 1.** Relative expression levels of *DA1*, *UBP12* and *UBP13* in the double *35S::GFP-DA1_35S::UBP12* and *35S::GFP-DA1_35S::UBP13* overexpression lines.

By limiting their peptidase activity, UBP12 and UBP13 fine-tune leaf growth, cell size and endoreduplication.

## Discussion

Ubiquitination is an important post-transcriptional modification that comes in various forms of complexity, which can lead to diverse effects on the fate of the substrate protein (*Callis, 2014*; *Swatek and Komander, 2016*). The latent peptidases DA1, DAR1 and DAR2 are activated upon multiple mono-ubiquitinations by BB and DA2 and can subsequently cleave several growth regulators (*Dong et al., 2017*). In this study, we identified two ubiquitin-specific proteases, UBP12 an UBP13, that interact with DA1, DAR1 and DAR2 in vitro and in vivo. Our experiments demonstrate that these UBPs work antagonistically to BB and DA2. Incubation of ubiquitinated DA1, DAR1 or DAR2 with either UBP12 or UBP13 resulted in strong deubiquitination. Interestingly, our in vitro immunoprecipitation (IP) shows that UBP12 and UBP13 can also interact with the unubiquitinated forms of DA1, DAR1 and DAR2. Similar deubiquitination experiments with other UBPs, such as UBP3, UBP15 and UBP24, demonstrated that this activity was specific for UBP12 and UBP13. Interestingly, *UBP15* has already been shown to genetically interact with *DA1* (*Du et al., 2014*) and UBP15 proteins can be cleaved by activated DA1 peptidases (*Dong et al., 2017*). Our results indicate that UBP15 has no deubiquitination activity towards DA1 and acts therefore downstream in this pathway. On the other hand, UBP12 and UBP13 are not substrates of DA1, unlike UBP15 (*Dong et al., 2017*; *Du et al., 2014*), which suggests they work more upstream in this signaling cascade. Most probably, UBP15 deubiquitinates and alters the fate of other downstream growth-regulating proteins. UBP12 and UBP13 have recently been described to deubiquitinate various poly-ubiquitinated proteins, preventing their proteasomal degradation (*An et al., 2018*; *Jeong et al., 2017*; *Lee et al., 2019*; *Park et al., 2019*). In addition, they can decrease the levels of mono-ubiquitinated H2A (*Derkacheva et al., 2016*) and remove multiple mono-ubiquitinations from DA1, DAR1 and DAR2 as described here. This demonstrates the flexibility of these UBPs towards different types of ubiquitination (*Clague et al., 2019*). Here, we could not find evidence that besides the removal of the multiple mono-ubiquitination sites of DA1, DAR1 and DAR2, potential degradation of these proteins by removal of poly-ubiquitination was mediated by UBP12 or UBP13.

In contrast to the early flowering time-related genes (*Cui et al., 2013*), downregulation of either *UBP12* or *UBP13* did not alter leaf size, indicating that these genes work redundantly in controlling leaf size. The leaf area was only reduced in *ubp12-2* mutants, in which levels of both *UBP12* and *UBP13* are decreased (*Cui et al., 2013*). The subsequently decreased deubiquitination activity towards DA1, DAR1 and DAR2 could result in an accumulation of the activated peptidases in *ubp12-2* mutants, leading to a decrease in cell number and leaf area. A similar phenotype is also observed in *GFP-DA1* overexpressing plants (*Vanhaeren et al., 2017*) and in the *ubp15* mutant (*Du et al., 2014*; *Liu et al., 2008*). In contrast, high ectopic expression of either *UBP12* or *UBP13* would result in very low levels of ubiquitinated DA1, DAR1 and DAR2, leading to a severe disturbance of leaf development. Indeed, *35S::UBP12* and *35S::UBP13* plants were strongly reduced in growth in a similar manner as *da1ko_dar1-1_dar2-1* triple mutants (*Peng et al., 2015*). A more detailed cellular and molecular analysis of *35S::UBP12* and *35S::UBP13* leaves revealed more parallels, such as a strong reduction in cell area and a decrease in ploidy levels at the early stages of leaf development, which were also observed in *da1ko_dar1-1_dar2-1* plants (*Peng et al., 2015*). The complete absence of DA1, DAR1 and DAR2 proteins in *da1ko_dar1-1_dar2-1* plants could however explain its stronger reduction in endoreduplication than that of *UBP12* and *UBP13* overexpression lines, in which ubiquitination of DA1, DAR1 and DAR2 can still occur, but is likely kept at a very low level. In addition, several markers of cell proliferation were found to be more highly expressed in *35S::UBP12* and *35S::UBP13* during early stages of leaf development, similarly as observed in plants that contain a DA1$^{R358K}$ mutation (*Vanhaeren et al., 2017*), which reduces the peptidase activity of DA1 (*Dong et al., 2017*). However, the final cell number in the *UBP12* and *UBP13* overexpression lines was not increased, suggesting that a strong reduction of DA1 activity rather delays development but does not increase the duration of cell proliferation and hence the final number of cells.

Previously, it has been shown that DA1, DAR1 and DAR2 negatively regulate their own activity by cleaving their activating E3-ligases BB and DA2, resulting in an activation-repression system (*Dong et al., 2017*). UBP12 and UBP13 form an additional layer of post-translational regulation to

limit the activity of DA1, DAR1 and DAR2 by specifically removing ubiquitin, and hence further fine-tune organ growth in this growth-regulatory pathway. This dual system to limit and fine-tune the activity of DA1, DAR1 and DAR2 demonstrates the importance of keeping a correct balance between the active and inactive pool of these proteases during leaf development. This is illustrated by the phenotype of plants in which this balance is altered. In wild-type conditions, a correct balance between an inactive and active (ubiquitinated) DA1, DAR1 and DAR2 results in an intact exit from mitosis and standard ploidy levels in leaf cells, which leads to normal plant growth (*Figure 7A*). High levels of UBP12 or UBP13 can however shift the balance to a depletion of activated DA1, DAR1 and DAR2, which results in a delayed endoreduplication, a severe reduction in cell size and stunted plant growth (*Figure 7B*), which is highly similar to the phenotype of *da1ko_dar1-1_dar2-1* mutants (*Peng et al., 2015*). Low levels of UBP12 and UBP13 might on the other hand lead to an increase in activated DA1, DAR1 and DAR2 levels, more destabilization of its substrates and can hence limit plant growth by reducing both cell area and cell number (*Figure 7C*). Higher levels of DA1 and mutations in *UBP15* have previously been reported to decrease leaf area and cell number (*Du et al., 2014*; *Liu et al., 2008*; *Vanhaeren et al., 2017*). In *ubp12-2* mutants and overexpression lines of *UBP12* and *UBP13*, leaf size and the average area of pavement cells are reduced. Because both very low and high levels of DA1 result in growth reduction (*Peng et al., 2015*; *Vanhaeren et al., 2017*), a tight balance between active and inactive DA1, DAR1 and DAR2 is crucial for normal plant development. With *UBP12* and *UBP13* expression levels being relatively constant throughout leaf development, unknown post-translational modifications of UBP12 and UBP13 are likely to alter the activity or specificity towards DA1, DAR1 and DAR2, as is the case with other deubiquitinating enzymes (*Huang and Cochran, 2013*). Additionally, it remains also unclear if this reduction in cell area in the *ubp12-2* mutants or *UBP12* and *UBP13* overexpression lines results only from altered levels of DA1 activity, or if UBP12 and UBP13 regulate cell expansion through other growth regulators.

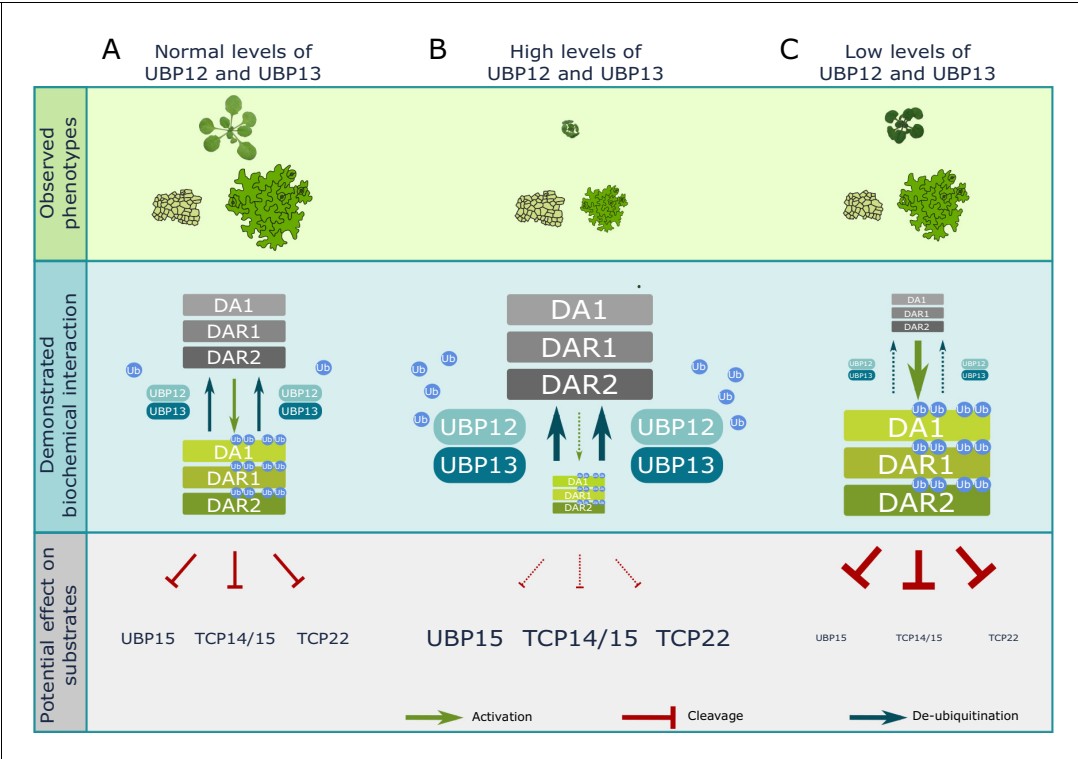

**Figure 7.** Model of UBP12 and UBP13 levels on leaf area and cellular phenotypes. Molecular balance and leaf phenotypes in (**A**) wild-type conditions, (**B**) high *UBP12* and *UBP13* expression lines and (**C**) lower levels of *UBP12* and *UBP13*.
The online version of this article includes the following figure supplement(s) for figure 7:

**Figure supplement 1.** Updated model of our current knowledge of the growth-regulating DA1 pathway.

Over the years, it has become increasingly clear that DUBs play a greater role in the development of eukaryotes than just processing and recycling free ubiquitin. Many DUBs are involved in fine-tuning molecular pathways by stabilizing proteins (*An et al., 2018*; *Jeong et al., 2017*; *Lee et al., 2019*; *Park et al., 2019*), through controlling protein endocytosis (*Crespo-Yàñez et al., 2018*; *Row et al., 2006*), by mediating DNA damage repair (*Nijman et al., 2005*) and by regulating transcription (*Derkacheva et al., 2016*; *Joo et al., 2007*; *Zhu et al., 2007*) to name a few. Potentially, DUBs could remove ubiquitin of existing chains so other chain types can be formed or other PTMs can be added.

Although our knowledge of organ growth has increased substantially over the last years (*Figure 7—figure supplement 1*), the developmental and environmental triggers that initiate the ubiquitination and deubiquitination of DA1, DAR1 and DAR2, the dominant-negative nature of DA1$^{R358K}$, additional potential substrates of the ubiquitin-activated peptidases and other post-translational regulatory elements remain some of the many compelling mysteries of this intriguing growth-regulatory pathway.

# Materials and methods

## Key resources table

| Reagent type (species) or resource | Designation | Source or reference | Identifiers | Additional information |
|---|---|---|---|---|
| Gene (*Arabidopsis thaliana*) | DA1 | TAIR | AT1G19270 | |
| Gene (*Arabidopsis thaliana*) | DAR1 | TAIR | AT4G36860 | |
| Gene (*Arabidopsis thaliana*) | DAR2 | TAIR | AT2G39830 | |
| Gene (*Arabidopsis thaliana*) | UBP12 | TAIR | AT5G06600 | |
| Gene (*Arabidopsis thaliana*) | UBP13 | TAIR | AT3G11910 | |
| Gene (*Arabidopsis thaliana*) | UBP3 | TAIR | AT4G39910 | |
| Gene (*Arabidopsis thaliana*) | UBP15 | TAIR | AT1G17110 | |
| Gene (*Arabidopsis thaliana*) | UBP24 | TAIR | AT4G30890 | |
| Gene (*Arabidopsis thaliana*) | DA2 | TAIR | AT1G78420 | |
| Cell line (*Arabidopsis thaliana*) | *Landsberg erecta* | TAIR/ABRC | Germplasm: 6530492727 NASC stock number: N84840 | https://www.arabidopsis.org/servlet/TairObject?id=4502009498&type=stock |
| Strain, strain background (*Escherichia coli*) | DH5 α | Thermo-fisher | 18258012 | Chemically competent cells |
| Strain, strain background (*Escherichia coli*) | BL21(DE3) | Thermo-fisher | EC0114 | Chemically competent cells |

## Plant material and growth conditions

All *Arabidopsis thaliana* mutants and overexpression lines that were used in this study were from the Col-0 background. *ubp12-1* (GABI_244E11), *ubp12-2* (GABI_742C10), *ubp13-1* (SALK_128312), *ubp13-2* (SALK_024054) and *ubp13-3* (SALK_132368) were kindly provided by Dr. Xia Cui (Chinese Academy of Agricultural Sciences, Beijing, China). The respective T-DNA insertions were verified by PCR. The *35S::GFP-DA1*, *35S::GFP-DAR1* and *35S::GFP-DAR2* lines were generated by Gateway cloning using the pK7WGF2 destination vector, the *35S::UBP12* and *35S::UBP13* lines using the

pFAST-G02. Because the latter overexpression lines contained the *OLE1::GFP* construct (*Shimada et al., 2010*), we could select positive seeds using a fluorescence binocular. The T-DNA lines were grown in soil for 25 days at 21°C and 16-h day/8-hr night cycles for the phenotyping experiments. All overexpression lines and the *da1-1* mutant were grown in vitro on plates containing half-strength Murashige and Skoog (MS) medium supplemented with 1% sucrose with a density of one plant per 4 cm$^2$. These plants were grown for 21 days at 21°C and 16-h day/8-hr night cycles. Seedlings for the immunoprecipitation followed by tandem mass spectrometry (IP-MS/MS) were grown for 8 days in liquid half-strength MS medium supplemented with 1% sucrose under shaking conditions (100 rpm) at 21°C and 16-h day/8-hr night cycles. For all experiments, the seeds were stratified in the dark for 2 days at 4°C before being placed in the respective growth rooms. Each quantitative experiment was performed in at least three independent biological repeats, meaning they were sown and harvested at a different time. All genotyping and cloning primers that were used in this study are listed in *Supplementary file 1*.

## Leaf measurements and cellular analysis

Leaves were dissected from the rosette and placed on a square plate containing 1% agar. The plants were imaged and the leaf area was analyzed using ImageJ v1.45 ((RRID:SCR_003070, NIH; http://rsb.info.nih.gov/ij/). For the cellular analysis, samples of leaf 3 (overexpression lines) and leaf 5 (*ubp12-2*) were cleared in 70% ethanol and mounted in lactic acid on a microscope slide. The total leaf blade area was measured for at least ten representative leaves under a dark-field binocular microscope. For mature leaves, abaxial epidermal cells at the center of the leaf blade, avoiding major veins, were drawn with a microscope equipped with differential interference contrast optics (DM LB with 403 and 633 objectives; Leica) and a drawing tube for at least three leaves. Photographs of leaves and scanned cell drawings were used to measure leaf and individual cell area, respectively, as described by *Andriankaja et al. (2012)*. The statistical analysis of the cellular data was performed in R 3.5.2 (www.r-project.org, RRID:SCR_001905). For the cellular profiling during the transition of cell proliferation to cell expansion, abaxial cells were drawn at the tip of the third leaf at 12 DAS. Scans of cellular drawings were analyzed using ImageJ v1.45 (RRID:SCR_003070, NIH; http://rsb.info.nih.gov/ij/) and cells were colored according to their area in Inkscape (https://inkscape.org/, RRID:SCR_014479).

## In vitro deubiquitination and cleaving assays

The coding sequences of *UBP3*, *UBP12*, *UBP13*, *UBP15* and *UBP24* were inserted in the pDEST15 (Thermo Fisher, RRID:SCR_008452) destination vector using Gateway cloning to generate GST-UBP3, GST-UBP12, GST- GST-UBP13, GST-UBP15 and GST-UBP24. The coding sequences of *DA1*, *DAR1* and *DAR2* were cloned into pDEST-HIS-MBP using Gateway cloning to generate HIS-MBP-DA1, HIS-MBP-DAR1 and HIS-MBP-DAR2. The HIS-DA2 (pET24a) construct was kindly provided by Prof. Michael Bevan (JIC, Norwich, UK). To generate the respective catalytic mutants, we performed site-directed mutagenesis on the entry clones of the respective genes by performing a PCR with primers containing the mutation. After the PCR, 5 µl of Buffer B and 1 µl of DpnI (Promega, RRID:SCR_006724) were added to each reaction. After an overnight incubation, competent DH5α *E. coli* cells were transformed and the presence of the mutation was checked by sequencing. All expression vectors were transformed into competent BL21 (DE3) *E. coli* cells. For each protein, the optimal conditions to obtain sufficient soluble proteins were determined (*Supplementary file 2*). GST-tagged proteins were purified from the bacterial lysate with Glutathione Sepharose 4B beads (GE Healthcare, 17075601) and HIS-tagged proteins with NI-NTA agarose beads (QIAGEN, 30210). Purified proteins were loaded on 4–15% Mini-PROTEAN TGX Precast Protein Gels (Biorad, 4561083DC), stained overnight with Instant Blue (Sigma-Aldrich, ISB1L-1L) and quantified using a BSA standard curve in Image Lab (RRID:SCR_014210, Biorad). The ubiquitination of DA, DAR1 and DAR2 and the cleaving assays were performed as described before (*Dong et al., 2017*). After the ubiquitination step, a 1:1 ratio of deubiquitinating enzymes to the ubiquitinated proteins were added and the reaction mix was incubated for 4 hr at 30°C. Reactions were stopped by adding 5x SDS sample buffer and boiled for ten min at 90°C. The samples were loaded on 4–15% or 7.5% Mini-PROTEAN TGX Precast Protein Gels (Biorad, 4561083DC). The proteins in the gels were transferred to a PVDF membrane using Trans-blot turbo transfer packs (Biorad, 170–4156) and the membranes were incubated

overnight in a 3% skimmed milk (Difco) 1x TBST solution. After blocking, GST-tagged proteins were detected with Anti-GST HRP Conjugate (Sigma-Aldrich, GERPN1236, RRID:AB_2827942) and MBP-tagged proteins with Anti-MBP Monoclonal Antibody (NEB, E8030S, RRID:AB_1559728) and subsequently with a secondary Rabbit IgG HRP Linked antibody (Sigma-Aldrich Cat# GENA934, RRID:AB_2722659). The antibodies were used following the manufacturer's instructions.

## RNA extraction, cDNA preparation and q-RT-PCR

Total RNA was extracted from flash-frozen seedlings or isolated leaves with TRIzol reagent (Invitrogen). Young seedlings (until 14 DAS) from which the leaves were isolated using a binocular were submerged overnight in RNA Later (Ambion) to prevent RNA degradation. To eliminate the residual genomic DNA present in the preparation, the RNA was treated by RQ1 RNAse-free DNase according to the manufacturer's instructions (Promega) and purified with the RNeasy Mini kit (Qiagen). Complementary DNA was made with the QScript cDNA supermix kit (Quantabio, 95048–100) according to the manufacturer's instructions. Q-RT-PCR was done on a LightCycler 480 (Roche) in 384-well plates with LightCycler 480 SYBR Green I Master mix (Roche) according to the manufacturer's instructions. Primers were designed with the Primer3 (RRID:SCR_003139, http://frodo.wi.mit.edu/) (*Supplementary file 1*). Data analysis was performed using the ΔΔCT method (*Pfaffl, 2001*), taking the primer efficiency into account. The expression data was normalized using three reference genes (AT1G13320, AT2G32170, and AT2G28390) according to the GeNorm algorithm (*Vandesompele et al., 2002*). The statistical analysis (ANOVA, Dunnett's test) was performed in GraphPad Prism 8.1 (www.graphpad.com, RRID:SCR_002798).

## Flow cytometry

The first leaf pair was harvested from 9 to 27 DAS with a three-day interval and frozen in liquid nitrogen. At least three leaves per time point of each biological repeat (n = 3) were chopped with a razor blade in 200 µL of Cystain UV Precise P Nuclei Extraction buffer (Sysmex), followed by the addition of 800 µL of Cystain UV Precise P staining buffer (Sysmex) and filtering through a 50-mm filter. Nuclei were analyzed with the Cyflow MB flow cytometer (Partec) and the FloMax software (RRID:SCR_014437). The statistical analysis (ANOVA, Dunnett's test) was performed in GraphPad Prism 8.1 (www.graphpad.com, RRID:SCR_002798).

## IP-MS/MS

The IP-MS/MS was based on the protocol from *Wendrich et al. (2017)*. For each pull-down, we used 3 g of homogenized Arabidopsis seedlings. The powder was dissolved in 4.5 ml BHB+ buffer supplemented with 4.5 µl benzonase and incubated for 30 min at 4°C. Then, the samples were further mixed three times for 30 s at 18,000 rpm using Ultra-TURRAX miniprobes (IKA). Subsequently, the mixtures were incubated for 30 min at 4°C on an end-over-end rotor. After incubation, the cellular debris was pelleted by two centrifugation steps at 14,000 rpm at 4°C in an Eppendorf centrifuge and further withheld by a 0.45-µm filter (Sartorius). The protein content was measured (OD 595) using a Bradford (Biorad) standard curve and equal amounts of proteins were incubated with 50 µl pre-washed anti-GFP-beads (µMacs, Miltenyi Biotec, RRID:AB_2827943) for 1 hr at 4°C in an end-over-end shaker. To isolate the beads, the columns were placed in the magnetic holder and washed four times with 200 µl BHB+ buffer and two times with 500 µl $NH_4HCO_3$ buffer. The purified proteins were eluted stepwise by 50 µl 95°C hot $NH_4HCO_3$ each time until no more beads pass through the column. Then, 4 µl Tryp/LysC mix (Promega) was added and the proteins were digested on-bead for 4 hr at 37°C with agitation (800 rpm) on an Eppendorf thermomixer. The digested mix was loaded again on the µMacs column attached to the magnetic holder to separate the eluate from beads. The eluate was collected in a new protein low binding Eppendorf tube and additionally 2 µl Tryp/LysC was added for an overnight digestion at 37°C with agitation (800 rpm) in an Eppendorf thermomixer. Finally, the samples were snap-frozen in liquid nitrogen and freeze-dried in a Speedvac (Labconco). Protein identification and data analysis were performed as described before (*Van Leene et al., 2015*).

### In vitro and in vivo pull-own

The in vitro pull-down experiments were performed as described before (*Dong et al., 2017*). As recombinant bait proteins, we used equal quantities of recombinant GST (Prospec, ENZ-393) and full-length GST-UBP12 or GST-UBP13. For the in vivo pull-down, full expression constructs of *35S:: RFP-DA1*, *35S::RFP-DAR1*, *35S::RFP-DAR2*, *35S::GFP-UBP12* and *35S::GFP* were amplified by PCR and cloned into Golden Gate modules (pGGA000, pGGB000) using the Gibson assembly method (NEB, E5510S). Existing *bsaI* sites in the Gateway p35S and DAR1 coding sequence were mutated by site-directed mutagenesis using *DpnI*. Subsequently, these constructs were cloned as interaction pairs on a Golden Gate vector to ensure the co-expression of the constructs in each cell. Arabidopsis cell suspension cultures were transformed as described before (*Van Leene et al., 2011*). Proteins were extracted and purified as described above and the purified fraction was subjected to Western blot. After blocking, GFP-tagged proteins were detected with anti-GFP (Abcam, ab290, RRID:AB_303395) and RFP-tagged proteins with anti-RFP antibody (Chromotek, RFP antibody [6G6], RRID: AB_2631395) and subsequently with a secondary Rabbit IgG HRP-linked antibody (Sigma-Aldrich, NA934v, RRID:AB_2722659) or secondary Mouse IgG HRP-linked antibody (Sigma-Aldrich, NA931v, RRID:AB_2827944), respectively.

### Cell free deubiquitination assay

Proteins were freshly extracted from 8-day-old seedlings of Col-0, *ubp12-2*, *35S::UBP12* an *35S:: UBP13* using a 0.5 M sucrose, 1 mM MgCl2, 10 mM EDTA (pH 8.0), 5 mM DTT, 50 mM Tris-MES (pH 8.0) extraction buffer without protease inhibitors (one µl of buffer was added per mg plant material). Subsequently, 400 µg of cell-free extract was added per 200 ng ubiquitinated HIS-MBP-DA1 proteins. For each time point, equal amounts of reaction mix were taken and the reaction was stopped by the addition of SDS sample buffer. The samples were blotted and detected as described above. To quantify the deubiquitination, a rectangle was drawn in ImageJ (RRID:SCR_003070, NIH; http:// rsb.info.nih.gov/ij/) spanning the size of fully ubiquitinated HIS-MBP-DA1. Then, the lane intensities were plotted and the pixels within the resulting curves were measured.

### Transient expression and confocal microscopy

For the co-localization experiments, the coding sequences of *DA1*, *DAR1* and *DAR2* were cloned in the gateway destination vector pK7WGR2 and those of *UBP12* and *UBP13* in pK7WGF2 to generate N-terminal RFP- and GFP fusion constructs, respectively. A 3-ml culture of competent LBA 4404 cells containing these constructs or P19 was grown overnight (28˚C) in YEB medium with the appropriate antibiotics. The next day, one ml of this culture was inoculated in a 9-ml YEB, 10-mM MES, 20-µM acetosyringone solution with the appropriate antibiotics and incubated overnight at 28˚C. Then, the bacteria were washed twice in a 100-mM MGCl2, 10-mM MES, 20-µM acetosyringone buffer and the $OD_{600}$ was adjusted to one for all cultures and further incubated for two hours at 28˚C. Then, equal mixtures of the constructs were infiltrated in 4-week-old *Nicotiana benthamiana* leaves. After three nights, leaf discs were imaged with a Zeiss LSM 710 confocal inverted microscope (RRID:SCR_018063, Zeiss) to image the RFP- (lasers: 561 nm: 3.0%, beam splitters: MBS: MBS 488/561, pinhole: 100 µm,, digital gain: 1.00, Master gain: 925) and GFP fusion proteins (lasers: 488 nm: 2.0%, beam splitters: MBS_InVis: Plate, pinhole: 100 µm, digital gain: 0.75, Master gain: 956).

## Acknowledgements

We like to express our gratitude to our colleagues of the Systems Biology of Yield and the Proteomics group, in particular Marieke Dubois and Ting Li, for all constructive discussions and scientific advice. We also like to thank Nancy Helderwert for preparing a tremendous amount of LB+ medium and Annick Bleys for her help in preparing the manuscript. The IP-MS/MS samples were processed by the VIB Proteomics Core (Ghent University, 9000 Ghent, Belgium).

## Additional information

### Funding

| Funder | Grant reference number | Author |
|--------|------------------------|--------|
| Bijzonder Onderzoeksfonds | BOF08/01M00408 | Mattias Vermeersch |
| Bijzonder Onderzoeksfonds | 01SC3117 | Ying Chen |
| China Scholarship Council | 201604910566 | Ying Chen |
| Research Foundation Flanders | 12V0218N | Hannes Vanhaeren |

The funders had no role in study design, data collection and interpretation, or the decision to submit the work for publication.

### Author contributions

Hannes Vanhaeren, Conceptualization, Data curation, Formal analysis, Funding acquisition, Validation, Investigation, Visualization, Methodology, Project administration; Ying Chen, Mattias Vermeersch, Valerie De Vleeschhauwer, Annelore Natran, Geert Persiau, Data curation, Investigation; Liesbeth De Milde, Data curation, Formal analysis, Investigation; Dominique Eeckhout, Data curation, Formal analysis; Geert De Jaeger, Resources, Formal analysis; Kris Gevaert, Dirk Inzé, Conceptualization, Supervision, Funding acquisition

### Author ORCIDs

Hannes Vanhaeren  https://orcid.org/0000-0003-3343-377X
Geert De Jaeger  https://orcid.org/0000-0001-6558-5669
Kris Gevaert  https://orcid.org/0000-0002-4237-0283
Dirk Inzé  https://orcid.org/0000-0002-3217-8407

### Decision letter and Author response

Decision letter https://doi.org/10.7554/eLife.52276.sa1
Author response https://doi.org/10.7554/eLife.52276.sa2

## Additional files

### Supplementary files

• Supplementary file 1. Primer list.
• Supplementary file 2. Conditions recombinant protein production.
• Transparent reporting form

### Data availability

All generated data is included in the data source files.

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
