## [Decision Letter]

**Acceptance summary:**

This work reveals the complex interplay of ubiquitin binding proteins (UBPs) and DA1-related proteins, shedding mechanistic light on ubiqutin-dependent cell identity switches and the control of organ size.

**Decision letter after peer review:**

Thank you for submitting your article "UBP12 and UBP13 negatively regulate the activity of the ubiquitin-dependent peptidases DA1, DAR1 and DAR2" for consideration by *eLife*. Your article has been reviewed by three peer reviewers, and the evaluation has been overseen by Jürgen Kleine-Vehn as the Reviewing Editor and Christian Hardtke as the Senior Editor. The reviewers have opted to remain anonymous.

The reviewers have discussed the reviews with one another and the Reviewing Editor has drafted this decision to help you prepare a revised submission.

The reviewers appreciated the mechanistic advance of your work, unraveling the regulatory importance of UBPs for the ubiquitin-dependent regulation of DA1, DAR1, and DAR2. However, they request that you should further consolidate your core findings prior publication at *eLife*. Mandatory requests include:

1) Please, provide a confirmation of the physical interaction of UBPs and DAs through another method.

2) Please, illustrate the differences in the level of DA1 ubiquitination depending on UBP12 and UBP13 activity.

3) Please, revise the model at the end so it only reflects what has definitely been shown in this (and previous) studies.

4) To complement the in vitro evidence, the reviewers kindly ask to possibly test whether overexpression or downregulation of UBP12/3 indeed affect DA1/DAR1/2 peptidase activity and thereby impact cleavage or protein stability of UBP15 and TCPs in protoplasts.

Please, also see the specific comments of the reviewers below, which you may also discuss in your revised manuscript.

Reviewer #1:

The manuscript by Vanhaeren et al. reports that UBP12/13 deubiquitinate DA1, DAR1 and DAR2, which might reduce their peptidase activity. Overexpression of UBP12 or UBP13 strongly decreased leaf size and cell area, and had lower ploidy levels. Mutations in UBP12 and UBP13 also produced smaller leaves that contained fewer and smaller cells. It seems interesting, but several conclusions are not justified.

1) If UBP12/13 can deactivate DA1/DAR1/2 (Figure 6), mutations in UBP12/13 could increase the accumulation of TCP15/22 and UBP15, resulting in larger and fewer cells. However, ubp12 *ubp13* mutants produced smaller leaves that contained smaller and fewer cells. In addition, the authors did not confirm whether ubp12 ubp13 could affect the levels of TCP15/22 and UBP15 proteins. Thus, the working model in Figure 6 might be incorrect.

2) It is unclear whether UBP12/13 can really remove monoubiquited DA1/DAR1/DAR2 or/and polyubiquitated DA1/DAR1/2 in vitro and in vivo. The polyubiquitination might affect protein stability. As the authors generated *35S:UBP12 / 35S:GFP-DA1* and *35S:UBP13 / 35S:GFP-DA1* plants, it is easy to test whether UBP12/13 could affect the stability of DA1 by Western blots.

3) If UBP12/13 can repress the activity of DA1/DAR1/2, it is necessary to test whether mutations in DA1/DAR1/2 could suppress the phenotypes of ubp12/13.

4) The authors showed the in vivo interaction of DA1 family proteins with UBP12 and UBP13 by label-free MS analyses. However, it's better to provide more interaction evidence by other approaches since MS analyses give false positive results some time.

Reviewer #2:

This paper by Vanhaeren et al. identifies and characterises a novel role for two deubiquitinating enzymes (UBP13 and 13) in the control of peptidase activity linked to cell growth. It was previously shown that the E3 ligases BB and DA2 monoubiquitinate and activate the latent peptidases DA1, DAR1 and DAR2, which then go on to limit leaf growth through disabling a range of positive regulators. Using a combination of mass-spec based screening, genetic manipulation, phenotypic analysis of growth and cell size, and in vitro biochemistry, the authors of this work identify UBP12 and 13 as interaction partners of the DA1/DAR1/DAR2 peptidases, and show convincingly that they act upstream of peptidase function through specific and selective deubiquitination. Thus, this work identifies a new level of regulation in the control of peptidase activity linked to cell growth, and provides new insights into how deubiquitination can play a major regulatory role in growth-associated cellular processes. Overall the experimental work is of a good standard, and the data presented support the conclusions drawn. The paper is well written, and the figures are clear.

Reviewer #3:

The manuscript by Vanhaeren and co-authors describes a potential regulatory link between Arabidopsis UBP12 and UBP13 deubiquitinases and the ubiquitin-dependent peptidases DA1, DAR1 and DAR2 (termed DAs hereafter) to fine tune leaf growth and development. It has been proven that monoubiquitination at multiple sites of DA proteins, mediated by E3 ubiquitin ligases BIG BROTHER and DA2, is crucial to activate the DA peptidases, which in turn will cleave key leaf growth regulators, such as UBP15, TCP14, TCP15 and TCP22. Therefore, by identifying a mechanism to counteract DA activation, based on DA deubiquitination, the authors provide new insights on the regulatory networks that modulate the duration of cell proliferation and the transition to endoreduplication and differentiation during leaf formation. In this context, their findings have additional clear implications for crop improvement.

However, despite the relevance and interest of these findings, a number of issues should be addressed to substantiate the model proposed by the authors.

First, whereas biochemical evidence is provided about UBP12 and UBP13 deubiquitinating activity towards DA proteins (Figure 4), in vivo data supporting these results is completely missing. The levels of ubiquitinated DA proteins should be analyzed in plants with altered expression of UBP12 and UBP13 (described in this manuscript).

Second, information about protein-protein interaction between UBP12 and UBP13 and DA proteins is limited to immunoprecipitation-MS data. Such results should be confirmed by additional means that also describe where, at the cellular and subcellular level, those interactions occur (nuclei?).

Third, UBP12 and UBP13 are known to deubiquitinate several, in principle unrelated, targets. However, it remains unknown which mechanisms provide them specificity to recognize a specific set of ubiquitinated targets (i.e. DAs) in order to modulate their activity. Thus, the authors should provide evidence on the precise environmental and developmental conditions, and the signaling mechanisms involved, that trigger UBP12 and UBP13 function towards DAs.

Fourth, reduced and increased activity of UBP12 and UBP13 in *ubp12-2* mutants and overexpression lines, respectively, lead to similar leaf and seedling phenotypes (Figure 2, Figure 2—figure supplement 8, summarized in Figure 6). Indeed, in the Discussion section (third paragraph) the authors mention that high levels of UBP12 and UBP13 lead to a severe reduction in cell size. On the other hand, low levels of UBP12 and UBP13 reduce cell area. This is counterintuitive. A more detailed discussion should be provided to better explain the outputs of UBP12 and UBP13 function in balancing DA protein activation/inactivation in order to control plant development.

Fifth, in the last paragraph of the subsection “Overexpression of UBP12 or UBP13 delays the onset of endoreduplication”, and Figure 3 and Figure 3—figure supplements 3 and 4, the authors show that overexpression of UBP12 and UBP13 induces expression, at certain time points, of several genes that positively regulate growth. This is contrary to the plant phenotype for those lines, where growth is reduced. These contradictory results should be discussed.

Sixth, Figure 4C. UBP12- and UBP13-mediated deubiquitination of ubiquitinated DAR2 is not clear at all, compared to that of ubiquitinated DA1 and DAR1. In addition, the fact that the shifted band with higher MW, corresponds indeed to ubiquitinated DA proteins should be proven (showing the in vitro DA2-mediated ubiquitination assays with appropriate controls).

---

## [Author Response]

The reviewers appreciated the mechanistic advance of your work, unraveling the regulatory importance of UBPs for the ubiquitin-dependent regulation of DA1, DAR1, and DAR2. However, they request that you should further consolidate your core findings prior publication at eLife. Mandatory requests include:1) Please, provide a confirmation of the physical interaction of UBPs and DAs through another method.

In the submitted manuscript, we had demonstrated the interaction between DA1, DAR1 and DAR2 with an in vivo IP-MS/MS, using Arabidopsis seedlings. This interaction was also confirmed by the in vitro deubiquitination assay. To further confirm these interactions, we performed an in vitro pull-down using recombinant GST, GST-UBP12 or GST-UBP13 as baits proteins, and HIS-MBP-DA1, HISMBP-DAR1 or HIS-MBP-DAR2 as prey proteins. After incubation and washing, our Western blots showed a strong interaction with GST-UBP12/13 and the prey proteins (HIS-MBP-DA1, HIS-MBP-DAR1 and HIS-MBP-DAR2), but not in the negative control (Figure 1D, E and F).

In addition, we performed an additional in vivo IP in Arabidopsis cell suspension cultures. For this purpose, full expression constructs of *35S::RFP-DA1, 35S::RFP-DAR1, 35S::RFP-DAR2, 35S::GFP-UBP12* and *35S::GFP* were amplified by PCR and cloned into golden gate modules using the Gibson assembly method. Existing *bsaI* sites in the Gateway p35S sequence and the *DAR1* coding sequence were mutated by site-directed mutagenesis using *dpnI* to make the constructs compatible for Golden Gate cloning. Subsequently, these constructs were cloned as interaction pairs on a Golden Gate vector to ensure the co-expression of the constructs in each cell. After an IP and WB, the interaction between GFP-UBP12 and RFP-DA1, RFP-DAR1 and RFP-DAR2 could be confirmed (Figure 1G).

The Materials and methods section has been updated.

2) Please, illustrate the differences in the level of DA1 ubiquitination depending on UBP12 and UBP13 activity.

In the original manuscript, in vitro evidence of the deubiquitination of the DA proteins by UBP12 and UBP13 was provided. in vivo in Arabidopsis, DA1 is a very lowly abundant protein which can only be visualized after IP ((Vanhaeren et al., 2017); Supplementary Figure 3D). In addition, in vivo ubiquitination of DA1 is very challenging to demonstrate. In the past, we extensively attempted to co-express GFP-DA1 with either BB or DA2, but we could never demonstrate an ubiquitination event by Western blotting. Only by adding high amounts of TR-TUBEs, synthetic proteins that specifically bind ubiquitin and might therefore stabilize this post-translational modification, to the extract of *35S::GFPDA1* during the extraction, it was possible to visualize ubiquitination patterns of GFP-DA1 in vivo (Dong et al., 2017). However, the process of adding TR-TUBEs to the extract followed by an immunoprecipitation, combined with the dramatic phenotypic differences between *35S::GFP_DA1* and *35S::GFP-DA1_35S::UBP12/13* plants might however induce a lot of variability. This could complicate a proper quantification of the deubiquitination of DA1. In addition, only limited seed material was available to perform these experiments due to fertility issues of the *35S::UBP12/13* lines, as is mentioned in the original manuscript.

Therefore, we applied an alternative approach based on well-established cell-free degradation approaches (Cheng et al., 2017; Dong et al., 2017; Garcia-Cano et al., 2014). Is such assays, fixed amounts of recombinant proteins are incubated with cell-free extracts of plants and their stability can be visualize over time by Western blot. Similarly, we started from exactly the same amount of ubiquitinated proteins that could be monitored over time. As starting material, newly prepared protein extracts (without protease inhibitors) of 8-day old seedlings of Col-0, *ubp12-2, 35S::UBP12* and *35S::UBP13*, that contained different levels of UBP12 and UBP13, were used. Subsequently, 400µg of cell-free extract was added per 200ng ubiquitinated HIS-MBP-DA1 proteins. From this reaction mix, equal amounts were taken per time point and deubiquitination was stopped by the addition of SDS-containing buffer. The samples were then subjected to Western blot and detection was done with anti-MBP. To quantify deubiquitination, a rectangle was drawn in ImageJ spanning the size of fully ubiquitinated HIS-MBP-DA1. Then, the lane intensities were plotted and the pixels within the resulting curves were measured.

In the Western blots shown in Figure 5A, we can clearly observe a rapid decrease in molecular weight of the HIS-MBP-DA1 proteins after addition of Col-0 extract (Figure 5A), which was not observed when *ubp12-2* extract was added (Figure 5B-C).

Next, we performed a similar experiment that also included the overexpression lines, which have high levels of UBP12 or UBP13. First, we could observe again that a slower rate of deubiquitination in samples where *ubp12-2* extract was added compared to those with Col-0 (Figure 5D-E). To be as stringent as possible, the stronger HIS-MBP-DA1 band of the *ubp12-2* Western blot at 0 h (Figure 5E, lane 7) was taken to calculate the deubiquitination (see Figure 5—source data 1: Calculation of in vivo deubiquitination). For the other blots, the average of the two 0 h lanes were taken. In contrast with the slower deubiquitination with *ubp12-2* extract, a faster loss of the ubiquitinated HIS-MBP-DA1 was observed when an extract of *35S::UBP12* or *35S::UBP13* plants was added compared to Col-0 (Figure 5F-H).

Taken together, these experiments demonstrate that the level of DA1 deubiquitination depends on different in planta levels of UBP12 and UBP13 and strengthen the validity of the in vitro experiments. These data were included as Figure 5 in the manuscript and an additional Source data file containing the plots and measurements was added.

3) Please, revise the model at the end so it only reflects what has definitely been shown in this (and previous) studies.

We have indicated in the model in a graphical manner what was demonstrated phenotypically (Figure 7, top panel) and molecularly (Figure 7, middle panel) in this study, and what the potential outcomes on the substrates of DA1 are, based on published work (Figure 7, bottom panel).

In addition, we added an updated model of the complete molecular mode of action of the entire pathway, adding the results from this manuscript to the knowledge from previous studies. This figure (in high resolution) was added as Figure 7—figure supplement 1 in the manuscript.

4) To complement the in vitro evidence, the reviewers kindly ask to possibly test whether overexpression or downregulation of UBP12/3 indeed affect DA1/DAR1/2 peptidase activity and thereby impact cleavage or protein stability of UBP15 and TCPs in protoplasts.

To address this challenging request, we aimed to generate an in vivo situation with high levels of ubiquitinated DA1, which could then be counteracted by either UBP12, UBP13 and their respective catalytically dead mutants. For this purpose, we generated the following expression constructs:

1) *35S::RFP-DA1* and *35S::DA2-FLAG* to generate ubiquitinated DA1

2) *p35S::GFP-UBP12* and *p35S::GFP-UBP13* as inhibitors of DA1 activity and *p35S::GFP-UBP12^C208S^* and *p35S::GFP-UBP13^C207S^* as the respective negative controls

3) *pUBP10::HA-TCP14*, *pUBP10::HA-TCP15, pUBP10::HA-TCP22, pUBP10::HA-UBP15* and *pUBP10::HA-BB* as cleaving substrates

The following setup was used (only one substrate is shown) to detect cleaving by DA1 and inhibition of cleaving by UBP12 or UBP13.

**Author response table 1. resptable1:** 

*35S::RFP-DA1*	*35S::DA2-FLAG*	*p35S::GFP-UBP12*	pUBQ10::HA-BB	Cleaving inhibition
*35S::RFP-DA1*	*35S::DA2-FLAG*	*p35S::GFP-UBP12^C208S^*	pUBQ10::HA-BB	Cleaving
*35S::RFP-DA1*	*35S::DA2-FLAG*	*p35S::GFP-UBP13*	pUBQ10::HA-BB	Cleaving inhibition
*35S::RFP-DA1*	*35S::DA2-FLAG*	*p35S::GFP-UBP13^C207S^*	pUBQ10::HA-BB	Cleaving

Next, protoplasts were transformed with these four constructs and the cleavage of the substrate was used as a readout of DA1 activity. Unfortunately and despite numerous attempts, we were unable to simultaneously express all four constructs in protoplasts of either BY-2 cells, Arabidopsis mesophyll cells or Arabidopsis cell suspension culture cells.

To ensure a better simultaneous transformation (and hence expression) and in an attempt to enhance the detection sensitivity, we then generated the following constructs:

1) *35S::HA-DA1* and *35S::DA2-FLAG* to generate ubiquitinated DA1

2) *pUBP10::UBP12-BFP* and *pUBP10::UBP13-BFP* as inhibitors of DA1 activity and *pUBP10::UBP12^C208S^–BFP* and *pUBP10::UBP13^C207S^–BFP* as the respective negative controls

3) *pUBP10::NLS-GFP-TCP14-mCherry*, *pUBP10::NLS-GFP-TCP15-mCherry, pUBP10::NLS-GFPTCP22-mCherry, pUBP10::NLS-GFP-BB-mCherry* as cleaving substrates

These constructs of the cleaving substrates would allow us to detect the cleaving and inhibition of cleaving with confocal microscopy by measuring the differences of FRET, and hence cleaving of the substrate, in the presence of UBP12/13 compared to their catalytic mutants.

Because a simultaneous expression of these constructs in each cell is required, these constructs were amplified by PCR and cloned into Golden Gate modules (pGGA000, pGGB000, pGGC000, pGGD000) using the Gibson assembly method (NEB, E5510S). The sequences were verified and existing *bsaI* sites in the gateway p35S and coding sequences were mutated by site-directed mutagenesis using *dpnI* to make the constructs compatible for Golden Gate cloning. Subsequently, these constructs were cloned as multiple expression modules in a small Golden Gate vector (pGG-AG-kan). A schematic representation of a set of these modules can be found in Author response table 2:

**Author response table 2. resptable2:** 

*35S::HA-DA1*	*35S::DA2-FLAG*	*pUBP10::UBP12-BFP*	*pUBP10::NLS-GFP-TCP14-mCherry*	Cleaving inhibition, FRET
*35S::HA-DA1*	*35S::DA2-FLAG*	*pUBP10::UBP12^C208S^–BFP*	*pUBP10::NLS-GFP-TCP14-mCherry*	Cleaving, loss of FRET
*35S::HA-DA1*	*35S::DA2-FLAG*	*pUBP10::UBP13-BFP*	*pUBP10::NLS-GFP-TCP14-mCherry*	Cleaving inhibition, FRET
*35S::HA-DA1*	*35S::DA2-FLAG*	*pUBP10::UBP13^C208S^–BFP*	*pUBP10::NLS-GFP-TCP14-mCherry*	Cleaving, loss of FRET

Unfortunately and despite a large amount of work using protoplasts of BY-2 cells and Arabidopsis cell suspension culture cells, no fluorescent signal could be observed in these cells (after 12, 24 or 48 hours).

Additional requests:

We performed additional experiments to address the additional comments and suggestions of the reviewers. The suggested textual changes were applied and size bars were added where necessary.

Please provide an image or a reference for the leaf roundness of da1-1 plants and plants overexpressing UBP12 or UBP13.

We have measured the height and width of Col-0, *da1-1, 35S::UBP12* and *35S::UBP13* leaves and calculated the leaf index. The leaf index is the ratio of leaf height to width and hence a measure of leaf shape (Tsukaya, 2002). We found that the leaf index of *da1-1, 35S::UBP12* and *35S::UBP13* leaves was consistently smaller than that of Col-0, meaning the leaves have a rounder shape.

These figures were included as Figure 2—figure supplements 1-2 in the manuscript, the raw data and statistical analysis are added as Figure 2—source data 2.

Please test if the stability of DA1 is affected by the presence of high levels of UBP12 and UBP13 in vivo by Western blot.

We have extracted proteins of *35S::GFP-DA1, 35S::GFP-DA1_35S::UBP12* and *35S::GFPDA1_35S::UBP13* plants and performed a purification with anti-GFP beads from equal protein inputs. After purification, we submitted the samples to Western blot and detected the abundance of GFP-DA1 proteins. We could not identify dramatic increases in protein abundance in the *35S::GFPDA1_35S::UBP12* and *35S::GFP-DA1_35S::UBP13* samples compared to those of *35S::GFP-DA1*, suggesting UBP12 and UBP13 do not prevent potential removal of polyubiquitin chains of DA1 and hereby prevent its degradation by the proteasome.

This figure was added as Figure 5—figure supplement 1 in the manuscript.

Please frame the new findings of UBP12 and UBP13 limiting the activity of DA1 with the additional activation-destruction system of the E3-ligases.

This dual control system is indeed important to restrict the levels of activated DA1. Because DA1 is a negative regulator of growth, these mechanisms are pivotal to keep the balance between active and inactive DA1 and coordinate developmental transitions without compromising growth. We have now addressed this more elaborate in the Discussion section.

Reduced and increased levels of UBP12 and UBP13 lead to a reduction in cell area and leaf size, this counterintuitive observation should be addressed in more detail.

We agree this observation might be counterintuitive. Parallels can be found however with triple mutants of DA1, DAR1 and DAR2 (*da1ko_dar1-1_dar2-1*; (Peng et al., 2015)) and overexpression lines of *GFP-DA1* (Vanhaeren et al., 2017). Both lines have contrasting levels of DA1, but produce smaller leaves. Our and previous studies suggest that a tightly controlled balance of activated DA1 is required for a correct developmental switch from cell proliferation to cell differentiation and endoreduplication. If too few ubiquitinated DA1 (DAR1, DAR2) is present, this developmental switch is delayed or severely disturbed and plants are smaller. If, on the other hand, the levels of activated DA1 are too high, too many downstream growth regulators might be cleaved, resulting in a reduction of cell division.

We further elaborated this in the Discussion section of the manuscript.

The authors show that overexpression of UBP12 and UBP13 induces expression, at certain time points, of several genes that positively regulate growth. This is contrary to the plant phenotype for those lines, where growth is reduced. These contradictory results should be discussed.

We agree that the higher levels of positive regulators of growth during the younger stages of leaf development of *35S::UBP12* and *35S::UBP13* plants are contradictory with their smaller phenotypes. Many genes that were selected are markers of cell proliferation and their ectopic expression indeed results in an increase in final cell number and leaf size. Most importantly, these genes also show a specific expression pattern during the transition of cell proliferation (high expression) to cell expansion (low expression). It is for the latter reason that these genes were selected. This allowed us to confirm our flow cytometry data, which showed that the leaf tissue of *35S::UBP12* and *35S::UBP13* plants was in a more proliferative stage (high expression of these marker genes and more 2C and 4C) than those of the Col-0 (lower expression of these marker genes and more 8C). We also made additional cell drawings of the tip of leaves during this developmental transition from cell proliferation to expansion (12 DAS, (Andriankaja et al., 2012)). We could observe that the cells in Col0 leaves clearly started expanding and differentiating, whereas the leaf tips of *35S::UBP12* or *35S::UBP13* plants contained almost exclusively undifferentiated cells (see Figure 3D). These results, together with our observation that cell number was not significantly altered in *UBP12* an *UBP13* overexpression lines, indicate that especially during the early developmental stages of leaf growth, endoreduplication and cell differentiation are delayed in *35S::UBP12* and *35S::UBP13* leaves.

We rephrased the choice for these Q-RT-PCR genes and elaborated the interpretation of the data more extensively in manuscript and added the cellular data to the Results section.

Please, describe where, at the cellular and subcellular level, the interaction between DA1, DAR1, DAR2 and UBP12, UBP13 occur.

We transiently expressed combinations of 35S::RFP-DA1, 35S::RFP-DAR1, 35S::RFP-DAR2 and 35S::GFP-UBP12, GFP-UBP13 in *Nicotiana benthamiana* leaves. Using confocal microscopy we could show that these proteins co-localize in the cytoplasm and in the nucleus (see Figure 7—figure supplements 5-10).